# Regulatory logic of endogenous RNAi in silencing *de novo* genomic conflicts

**Jeffrey Vedanayagam[1], Ching-Jung Lin[1,2], Ranjith Papareddy[3], Michael Nodine[3¤], Alex S. Flynt[4], Jiayu Wen[5], Eric C. Lai[1]***

**1** Developmental Biology Program, Sloan Kettering Institute, New York, New York, United States of America, **2** Weill Graduate School of Medical Sciences, Weill Cornell Medical College, New York, New York, United States of America, **3** Gregor Mendel Institute (GMI), Austrian Academy of Sciences, Vienna Biocenter (VBC), Austria, **4** Cellular and Molecular Biology, University of Southern Mississippi, Hattiesburg, Mississippi, United States of America, **5** Division of Genome Sciences and Cancer, The John Curtin School of Medical Research The Australian National University, Canberra, Australia

¤ Current address: Cluster of Plant Developmental Biology, Laboratory of Molecular Biology, Wageningen University & Research, Wageningen, the Netherlands

* laie@mskcc.org

**Data Availability Statement:** RNA-seq, small RNA, 5'-seq, and 3'-seq data from D. melanogaster and D. simulans generated for this study were deposited in the GEO database: GSE230111. In

## Abstract

Although the biological utilities of endogenous RNAi (endo-RNAi) have been largely elusive, recent studies reveal its critical role in the non-model fruitfly *Drosophila simulans* to suppress selfish genes, whose unchecked activities can severely impair spermatogenesis. In particular, hairpin RNA (hpRNA) loci generate endo-siRNAs that suppress evolutionary novel, X-linked, meiotic drive loci. The consequences of deleting even a single hpRNA (*Nmy*) in males are profound, as such individuals are nearly incapable of siring male progeny. Here, comparative genomic analyses of *D. simulans* and *D. melanogaster* mutants of the core RNAi factor *dcr-2* reveal a substantially expanded network of recently-emerged hpRNA-target interactions in the former species. The *de novo* hpRNA regulatory network in *D. simulans* provides insight into molecular strategies that underlie hpRNA emergence and their potential roles in sex chromosome conflict. In particular, our data support the existence of ongoing rapid evolution of Nmy/Dox-related networks, and recurrent targeting of testis HMG-box loci by hpRNAs. Importantly, the impact of the endo-RNAi network on gene expression flips the convention for regulatory networks, since we observe strong derepression of targets of the youngest hpRNAs, but only mild effects on the targets of the oldest hpRNAs. These data suggest that endo-RNAi are especially critical during incipient stages of intrinsic sex chromosome conflicts, and that continual cycles of distortion and resolution may contribute to speciation.

## Author summary

Selfish meiotic drive loci promote their own transmission at the cost of host reproductive fitness. Therefore, their existence and activity places strong pressure to innovate genetic suppressors that can return gametogenesis to a normal Mendelian state. Because such genetic battles are typically kept in a silenced, cryptic state, the identity of underlying

addition, for D. simulans w[XD1] RNA-seq and small RNA data analyzed in this study were previously reported at doi: 10.1038/s41559-021-01592-z and available under GEO database ID: GSE185361.

**Funding:** JV was supported by a Pathway to Independence Award (NIH-K99GM137077). Research in MN's group was supported by the European Research Council under the European Union's Horizon 2020 Research and Innovation Program grant 637888. AF was supported by NSF 1845978, and Mississippi INBRE Institutional Development Award (IDeA) from the National Institute of General Medical Sciences of the National Institutes of Health (P20GM103476). JW was supported by an Australian Research Council (ARC) Future Fellowship (FT16010043) and an Australian National University (ANU) Futures Scheme. Work in ECL's group was supported by the National Institutes of Health (R01-HD108914 and R01-GM083300), BSF-2015398, and MSK Core Grant P30-CA008748. The funders had no role in study design, data collection and analysis, decision to publish, or preparation of the manuscript.

**Competing interests:** The authors have declared that no competing interests exist.

meiotic drivers and suppressors is often hidden. We find that endo-siRNAs generated by the hairpin RNA (hpRNA) substrates of the RNA interference (RNAi) pathway evolve rapidly, and strongly repress specific genes in the male germline that exhibit signatures of genetic conflict. This indicates that genetic analysis of RNAi-suppressed target networks is a new forward strategy to uncover loci with candidate meiotic drivers. More generally, these data flip the convention for gene regulatory strategies for the paralogous miRNA pathway. Whereas only the oldest miRNAs typically have gained high expression levels and substantial target repression, it is the very youngest hpRNA-siRNA loci that exhibit highest expression and most overt effects on gene silencing.

## Introduction

In sexually reproducing organisms, structurally distinct sex chromosomes (X/Y or Z/W) are involved in sex-specific regulatory processes, such as sex determination and dosage compensation [1, 2]. The genomic distinction of sex chromosomes, compared to their autosomal counterparts, underlies strikingly contrasting features including (1) reduction or lack of recombination, (2) strategies to equalize gene expression from the X or Z of males and females and (3) accumulation of repeats on the degenerating Y (or W) chromosome [3]. Accordingly, XY and ZW chromosomes are especially evolutionarily dynamic [4].

The rapid and continual evolution and emergence of sex chromosomes, along with their contrasting biological interests and fates, is linked to their involvement in intragenomic conflict [5] and sex chromosome meiotic drive [2]. In particular, selfish sex-linked genes can impair transmission of the reciprocal sex chromosome, thereby favoring the driving chromosome amongst progeny. Sex chromosome meiotic drive can be easily observed in deviation of *sex-ratio* (SR) from equality [6]. Fisher's principle proposes that, if males and females cost equal amounts to produce, an equal ratio of the sexes will be the equilibrium [7]. However, SR drive systems have been widely documented, indicating recurrence of sex chromosome drive in nature. Fisher proposed that sex-biased populations direct their reproductive efforts disproportionately to the rarer sex, thus tending towards normalization of SR over the long term. However, the molecular bases of SR distortion and restoration of parity are poorly understood. This is due in part due to the fact that, despite their ubiquity in nature, many well-studied model organisms lack documented SR drive systems. For example, even though mutants of sex determination or dosage compensation systems can distort the sex of viable progeny, a long history of genetic studies in the well-studied fruitfly *D. melanogaster* has not uncovered strong, selfish SR drive loci.

Curiously then, a history of genetic analyses uncovered three independent sex chromosome drive systems in *D. simulans*, a close sister species of *D. melanogaster* [8]. *D. simulans* bears the Winters, Durham, and Paris systems, meiotic drive systems that map to distinct genomic intervals and indicate multiple newly-emerged strategies that deplete male progeny [9–12]. Despite progress on the identification of potential drivers and/or suppressors for the three SR drive systems [9, 11, 13–16], much remains to be understood regarding their molecular mechanisms, and even whether SR meiotic drive loci have been comprehensively identified in this species. The limited genetic tools and genomic data in this non-model fruitfly have impeded efforts, even though *D. simulans* is arguably the premier model to explore the molecular bases of SR drive. Interestingly, our prior findings linked the *Winters* and *Durham* drivers to sperm chromatin packaging factor and HMG-box factor Protamine, providing clues to the evolutionary origins and mechanistic workings of these selfish genes [15, 16]. Indeed, Protamine-encoding

loci have duplicated recurrently across the Drosophilid phylogeny, and emerging evidence indicates, their rapid evolution and turnover may broadly be linked to meiotic drive in other lineages {Chang and Malik eLIFE 2023}. Moreover, recent work suggests that Y-linked multicopy members of the Mst77Y family, derived from autosomal protamine-like factor Mst77F, may act in a dominant-negative manner to preferentially decompact X sperm [17]. Although it is not yet clear that Mst77Y factors can explicitly distort progeny sex ratio, its *D. melanogaster*-specific amplification on the Y is suggestive of sex chromosome conflict.

Recently, we revealed that two genetically identified loci that suppress SR drive encode hairpin RNA (hpRNAs), which generate endogenous siRNAs (endo-siRNAs). In particular, the Winters SR suppressor (*Nmy*) and the Durham SR suppressor (*Tmy*) encode related hpRNAs that have capacities to silence the SR distorter *Dox* and its paralog *MDox* [14], to equalize SR in *D. simulans*. Notably, both *Dox* and *MDox* are located on the X chromosome and are silenced by endo-siRNAs in *D. simulans* testis. These attributes fit the proposition that meiotic drivers may preferentially be encoded on the X, and exploit male gametogenesis to gain unfair transmission advantages. Moreover, the family of Dox-related genes and related hpRNAs has undergone massive amplification in the *simulans*-clade sister species *D. sechellia* and *D. mauritiana*, but none of these driver or suppressor loci are present in the *D. melanogaster* genome [15, 16]. These findings are testament to the rapid evolution (both emergence and disappearance) of SR meiotic drive systems, and a key role for endo-RNAi to suppress incipient selfish genes located on the X.

To test for broader roles of RNAi in suppressing meiotic drive and/or sex chromosome conflict, we used short and long transcriptome data from mutants of the core RNAi factor *dcr-2* to perform a functional evolutionary comparison of hpRNA regulatory networks in *D. melanogaster* and *D. simulans* testis. We reveal asymmetric proliferation of evolutionarily novel hpRNAs in *D. simulans*, which preferentially repress *de novo* X-linked genes. This suggests broader roles for RNAi in taming sex chromosome conflict in this species. These loci also provide insights into the earliest molecular steps in the birth of hpRNAs. Surprisingly, the newest hpRNA-target interactions mediate much larger regulatory effects than the oldest hpRNA-target interactions, thereby inverting the convention of miRNA-mediated networks. Overall, we conclude that RNAi has a much larger role in silencing sex chromosome conflict than anticipated, and suggests that resolution of active intragenomic conflicts may contribute to speciation.

## Results

### Generation of *D. simulans dcr-2* deletion mutants marked by *white*[+]

We recently reported deletion alleles of core RNAi factors [*dcr-2* and *ago2*] in *D. simulans* [14]. Although these mutants are viable, they are completely male sterile, and therefore cannot be maintained as stable stocks. This presents a technical challenge since *D. simulans* lacks balancer chromosomes and the *3xP3:DsRed* marker was not fully reliable to distinguish heterozygotes from homozygotes. Therefore, preparation of pure homozygous material required extensive genotyping of small batches of dissected flies prior to combining samples for RNA isolation. Moreover, in initial RNA-seq analyses, genotyped samples were still prone to contamination. As a further complication, due to extremely high expression of many accessory gland transcripts [18], we noticed that even minute quantities of contaminating accessory gland could produce large biases in gene expression between libraries. Since *D. simulans* mutants were generated in a *white* mutant background (*w[XD1]*), their testis was colorless and thus more challenging to visualize compared to a *white+* background, where the testis is bright yellow. For these reasons, the preparation of suitable quantities of *D. simulans* RNAi mutant testis was not straightforward.

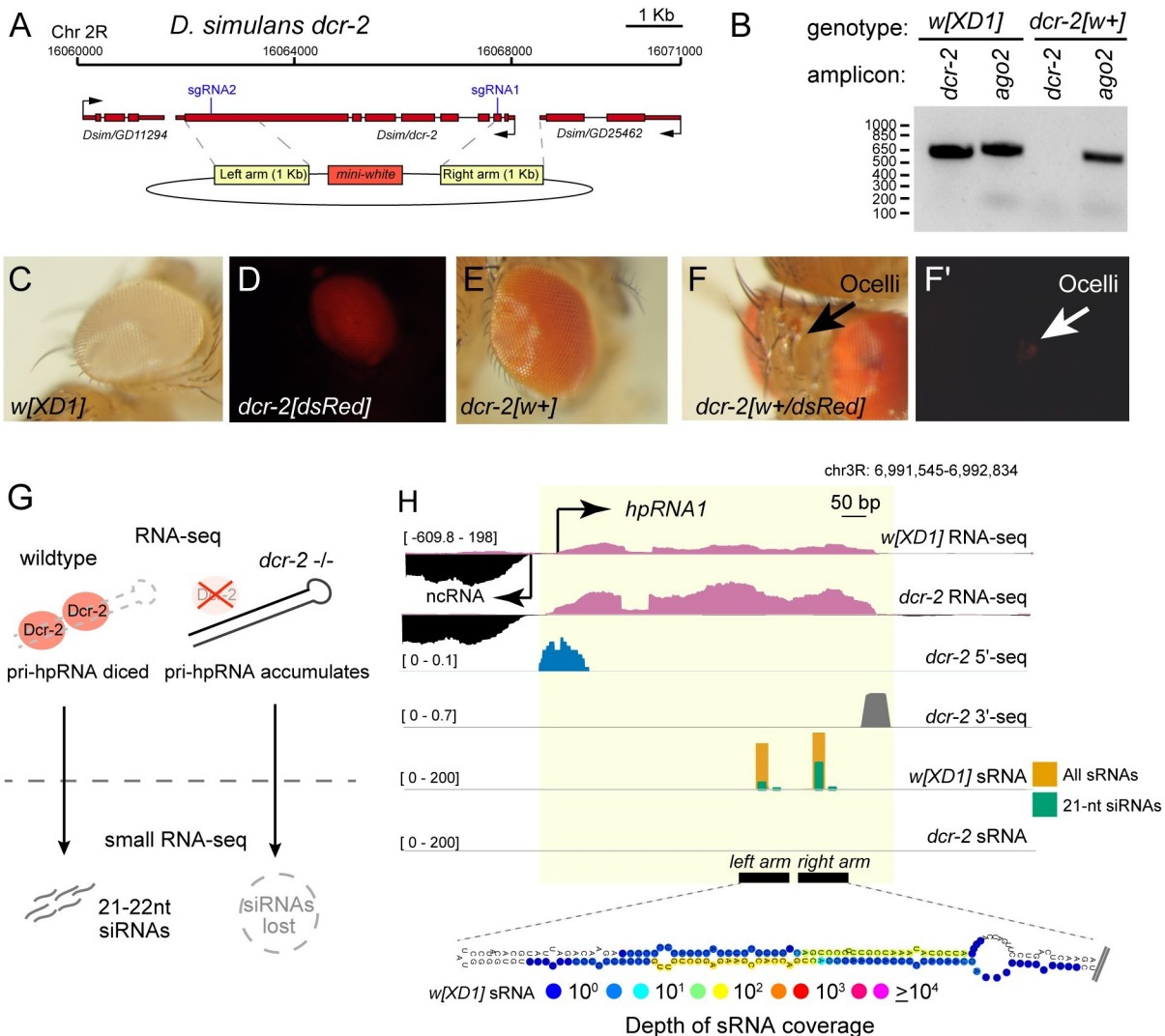

**Fig 1. Functional annotation of hpRNAs using small RNA and RNA-seq data *dcr-2* mutants.** (A) *dcr-2* location, sgRNAs, homology donor arms with white[+] marker used for mutant selection. (B) PCR genotyping and validation of *dcr-2[w+]* CRISPR mutant showing absence of *dcr-2* amplicon; *ago2* amplicon was used as control. (C) *D. simulans w[XD1]*, a *white* mutant background used for CRISPR. (D) *dcr-2[DsRed]* deletion allele exhibits red fluorescent eyes. (E) *dcr-2[w+]* deletion allele exhibits pigmented eyes. (F) *dcr-2[w+/DsRed]* trans-heterozygous mutants. DsRed+ eyes are not very visible in a *w+* background (F), but their fluorescence can be identified in the ocelli (F'), which are primitive light sensing organs. (G) Schematic of Dcr-2 processing of primary hpRNA (pri-hpRNA) transcripts into 21–22 nt siRNAs. pri-hpRNAs can be detected using RNA-seq, and the corresponding siRNAs derived from the hairpin can be identified via small RNA-seq. (H) *hpRNA1* illustrates the behavior of hpRNA-derived testis RNA products in wildtype and *dcr-2* mutants. Normalized RNA-seq and spike-in normalized sRNA-seq tracks of the hpRNA shown in *w[XD1]* and *dcr-2*. The top two tracks show RNA-seq of primary hpRNA transcript is increased in *dcr-2* mutants due to upregulation of hpRNA primary transcript. The middle tracks show that the 5' and 3' ends of *pri-hpRNA1* are defined by 5'-seq and 3'-seq data, respectively. The bottom two tracks show that *hpRNA1*-derived small RNAs are biased to 21 nt and mostly eliminated in *dcr-2* mutant.

To address these issues, we used CRISPR/Cas9 to generate multiple founders of a new *D. simulans Δdcr-2* null allele, where most of its coding region is replaced with *mini-white⁺* ($w^+$) (**Fig 1A and 1B**). We anticipated selecting homozygotes with deeper eye color, as is typical in *D. melanogaster*; however, this was also not fully reliable due to the red eyes of these alleles. Instead, by crossing *dcr-2* alleles marked by *3xP3:DsRed* and $w^+$ (**Fig 1C–1E**), we could select trans-heterozygotes carrying both dominant markers. Although DsRed⁺ eyes cannot be

effectively scored in a $w^+$ background, it is still possible to score DsRed$^+$ ocelli (**Fig 1F–1F'**). Our independent Δ*dcr-2[w⁺]* alleles were viable but specifically sterile in males, exhibited severe spermatogenesis defects, and failed to complement their corresponding DsRed alleles. We therefore used the trans-heterozygotes for subsequent analyses.

## Signature features of hpRNA loci in short/long RNAs from wildtype and *dcr-2* mutants

In contrast to *D. melanogaster* RNAi mutants, which are viable and sub-fertile [19], deletion of core RNAi factors in *D. simulans* result in complete male sterility [14]. This is due at least in part to the requirements of *Nmy* and *Tmy*, which are *de novo* hpRNAs that silence incipient X chromosome *sex ratio* distorters (*Dox* and *MDox*) in the male germline [14]. To assess the impact of RNAi loss more globally, and to compare *D. melanogaster* and *D. simulans* in greater detail, we generated biological replicates of small RNA and total RNA sequencing data from testis of *dcr-2* heterozygotes and mutants in *D. melanogaster*, and *w[XD1]* vs. *dcr-2* mutants in *D. simulans*. In control, we expect that primary hpRNA (pri-hpRNA) transcripts are cleaved by Dcr-2 into 21–22 nucleotide (nt) siRNAs, while *bona fide* siRNAs should be lost in *dcr-2* mutants concomitant with accumulation of their progenitor mRNAs (**Figs 1G and S1**). Together, this combination of datasets permits functional categorization of genuine hpRNAs with high specificity, as illustrated by *hpRNA1* (**Fig 1H**) and others (**S2 Fig**).

We plotted differential expression of *D. melanogaster* and *D. simulans* RNA-seq data in MA plots (**Fig 2A–2D**), which display the log-ratio of *dcr-2 vs.* wild type controls of RNA-expression (M-axis) and the average gene expression in the RNA-seq data (A-axis). Strikingly, on the genomewide scale, hpRNA precursors were amongst the highest-upregulated transcripts in *dcr-2* mutants (**Fig 2**). We initially assessed this using the set of *D. melanogaster* hpRNAs, all of which are conserved in *D. simulans* [19]. The known hpRNAs dominate the highest-upregulated transcripts in *dcr-2* mutants of both species. For example, *Dmel-hp-mir-997-1* was the 2nd-highest elevated locus genome-wide, and 7/9 hpRNAs were in the top 20 upregulated loci in *D. melanogaster* (**Fig 2A and 2B**). We similarly observed that pri-hpRNAs of *D. simulans* orthologs of *D. melanogaster* hpRNAs were strongly elevated in *dcr-2* mutants (**Fig 2C and 2D**). These effects were specific, since primary miRNA transcripts were largely unaffected in *dcr-2* mutants of either species (**Fig 2B and 2D**). This was expected, as miRNAs are processed instead by Dcr-1. One exception was *mir-985*, whose primary transcript was elevated in mutants of *D. melanogaster dcr-2* (**Fig 2A and 2B**), but not in *D. simulans dcr-2* (**Fig 2C and 2D**). The reason for this discrepancy is unknown, but a potential explanation is that transcription of *mir-985* is elevated in *D. melanogaster* as a secondary effect that is not shared in *D. simulans*. We also note elevated novel hpRNAs in *D. simulans* than in *D. melanogaster* (Fig 2C and 2D; two-tailed Fisher's exact test *P* = 0.0010). Finally, hpRNA-derived siRNAs were strongly depleted in *dcr-2* mutants, including from all previously classified *D. melanogaster* hpRNAs and their *D. simulans* orthologs (**Fig 2E and 2F**).

Inspection of the local genomic regions of known hpRNAs revealed provocative differences between *D. simulans* and *D. melanogaster* in the vicinity of hpRNA clusters, and provided additional evidence for rapid flux in hpRNA loci. The largest set of dispersed hpRNA loci in *D. melanogaster* are members of the *hp-pncr009* family, for which 3 separate hpRNAs (*hp-pncr009*, *hp-CR32207*, and *hp-CR32205*) are interspersed with 9 protein-coding target genes of the *825-Oak* family [19]. We have theorized that transcription across pairs of divergently-oriented *825-Oak* family loci might beget *pncr009* hpRNAs. Interestingly, in the short evolutionary distance that separates *D. melanogaster* and *D. simulans*, we identify two additional *pncr009*-class hpRNAs in *D. simulans* (**S3A Fig**). To facilitate intuitive connection of these

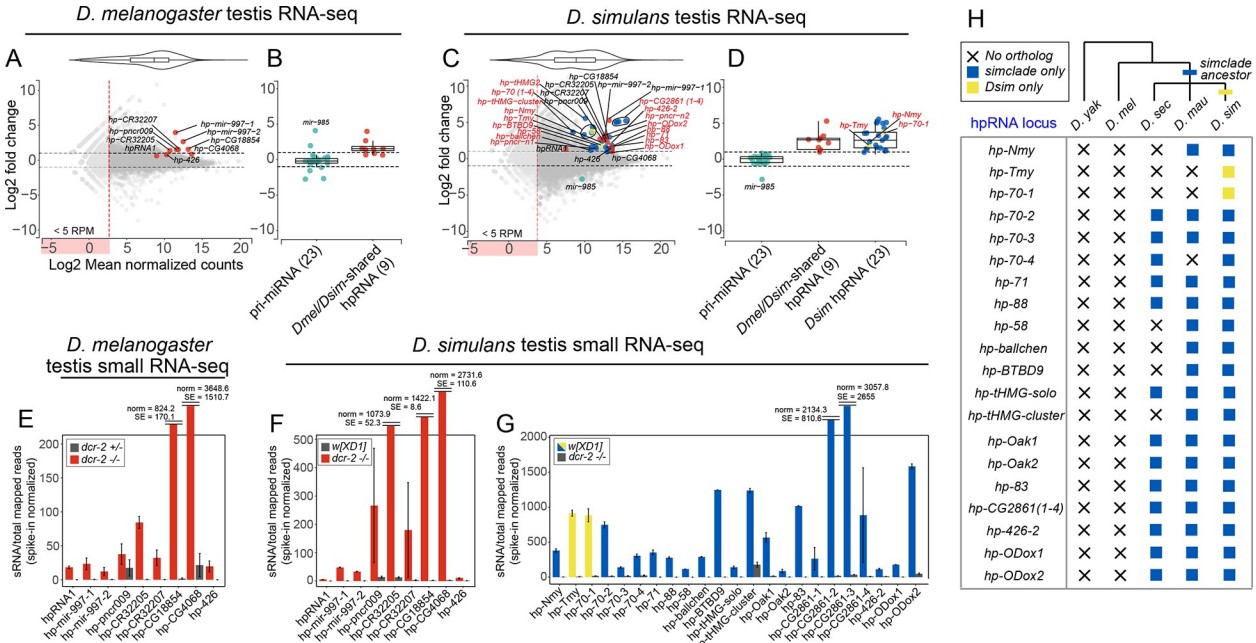

**Fig 2. Comparative hpRNA annotation using functional genomic data.** (A) MA-plot comparing *D. melanogaster* wild-type and *dcr-2* testis RNA-seq data. Red dots denote hpRNAs conserved with *D. simulans*, including one newly-recognized locus *hp-426*. (B) Primary hpRNA transcripts are all elevated in *D. melanogaster dcr-2* testis, unlike primary miRNA transcripts; *mir-985* is an exception. (C) MA-plot comparing *D. simulans* control *w[XD1]* and *dcr-2* testis RNA-seq data. Red dots mark hpRNAs conserved with *D. melanogaster*, while blue dots indicate *de novo* hpRNAs in *D. simulans*. (D). Comparison of primary hpRNA and primary miRNA transcripts in *D. simulans*. All pri-miRNA transcripts are unchanged (including *mir-985*), while all pri-hpRNA transcripts are elevated in both conserved and de novo hpRNAs. Compared to *D. melanogaster*, *D. simulans* has an elevated number of novel hpRNAs (two-tailed Fisher's exact test, *p* = 0.0010). (E-G) Expression of hpRNA-derived small RNAs in *D. melanogaster* (E) and *D. simulans* (F-G). Note that reads from the multicopy tandem hp-CG4068 repeats are condensed into a single hpRNA locus, to avoid over-tabulation of hpRNA numbers. All hpRNA-siRNAs are decreased in *dcr-2* mutants, in both species and regardless of the hpRNA age. (H) Phylogenetic analyses of the presence/absence of orthologs to newly-annotated *D. simulans* hpRNAs in the *simulans* clade sister species (*D. sechellia* and *D. mauritiana*), compared with *D. melanogaster* and the close outgroup species *D. yakuba*. All of the newly-annotated *D. simulans* hpRNAs were born within the *simulans* clade ancestor, or within *D. simulans* itself.

hpRNAs to target genes in the *825-Oak* family, we named these novel *D. simulans* hpRNAs as *hp-Oak1* and *hp-Oak2* (**Fig 2C and 2D**).

We also documented evolutionary flux in the tandem hpRNA repeats of the *hp-CG4068* cluster. Although the copy number was potentially in question from prior short-read genome assemblies, the recent availability of *simulans*-clade PacBio genomes [20] demonstrates radical copy number of the *hp-CG4068* cluster. There are 20 tandem copies in *D. melanogaster* but only 9 tandem copies in *D. simulans* (**S3B Fig**), as well as 14 copies in *D. mauritiana*, and 10 copies in *D. sechellia*. We note that the available data do not rule out that their copy numbers might not also be variable within a species. In any case, there is high evolutionary divergence in the copy number of hpRNAs located in both genomically linked copies (*hp-pncr009* cluster) as well as in tandem copies produced from a common transcript (*hp-CG4068* cluster). Such dynamics are much greater than observed for canonical miRNAs, which only occasionally exhibit similar changes amongst these species [21, 22].

## Unidirectional expansion of hpRNAs in *D. simulans* compared to *D. melanogaster*

With these validations of known hpRNAs in hand, we undertook more systematic hpRNA annotations as a basis of more comprehensive evolutionary comparisons. In particular, we

sought highly structured loci that produce ~21 nt-biased, Dcr-2-dependent small RNAs, that also accumulate primary transcripts in *dcr-2* mutants (**Fig 1H**). However, we did not absolutely require pri-hpRNA changes in *dcr-2*, for several reasons. First, pri-hpRNA transcripts might not be stable and/or might be subject to other RNA decay pathways. Taking the canonical miRNA pathway as an analogy, not all pri-miRNA transcripts accumulate as unprocessed full-length mRNAs in mutants of nuclear miRNA processing factor Drosha, and RNase III enzyme. Second, it was conceivable that some loci are processed earlier in development to yield stable small RNAs, but are not substantially transcribed in adult testis. Finally, transcription of some pri-hpRNA loci might be decreased in RNAi mutants.

In *D. melanogaster*, we previously annotated hpRNAs from nearly 300 small RNA libraries of diverse developmental, tissue and cell origins, yielding only nine confident hpRNAs [19]. Given that testis is the predominant location of hpRNA-siRNA accumulation, we interrogated our new genetically paired testis data for evidence of additional Dcr-2-dependent inverted repeat loci. However, beyond previously known hpRNAs, we only recovered a single new hpRNA, *hp426* (**Fig 2A, 2B, and 2E**), which produces siRNAs in *D. simulans*. Thus, we did not identify any *D. melanogaster*-specific hpRNAs.

A very different picture emerged from analysis of *D. simulans*. Although we had far fewer small RNA libraries to annotate from, especially of testis datasets, we found numerous novel hpRNAs (**S1 Table**). These annotations were of high confidence, as nearly all of them exhibited reciprocal behavior of primary transcripts and mature siRNAs, when comparing control and *dcr-2* testis libraries (**Fig 2C, 2D and 2G**). In fact, most novel pri-hpRNA loci resided amongst the most-upregulated transcripts across *D. simulans dcr-2* testis (**Fig 2C**). A majority of these accumulated discrete spliced RNAs. However, depending on the locus, transcript coverage was often non-uniform. This was particularly the case within highly-duplexed portions of foldback structures (**S4 Fig**), evidently indicating that library construction was compromised within strongly double-stranded transcript regions. Coverage was also an issue at transcript termini, which is generally the case with typical RNA-seq protocols [23].

We therefore employed two more approaches to help annotate pri-hpRNA transcripts in control and *dcr-2* mutant testis: analysis of 5' ends from low-input RNA [24] and 3'-end sequencing to determine polyadenylation sites [25]. As illustrated in **Fig 1I**, 5'-seq and 3'-seq directly visualize the beginnings and ends of pri-hpRNA transcripts, and indicate that hpRNA progenitors are mRNAs. We note that some hpRNA loci are exact genomic copies that remain valid upon scrutiny of the largely contiguous PacBio *D. simulans* assembly [20], a phenomenon we return to later in this study. However, even when conservatively counting identical hpRNA copies within a tandem cluster as a single locus, there are 23 distinct, *de novo* hpRNA transcription units in *D. simulans*, which can be assigned to 15 families that are not simply genomic copies (**Fig 2C, 2D, 2G, and 2H**). As described later, some of these families can still be traced as sharing evolutionary heritage (as is the case for *hp-Nmy*/*hp-Tmy*, which are related in sequence but have separable functions).

From head-to-head comparison of *D. melanogaster* and *D. simulans* alone, we cannot distinguish if *D. simulans* gained hpRNAs, or if *D. melanogaster* lost them. Therefore, we analyze the distribution of our newly annotated *D. simulans* hpRNAs in the sister *simulans*-clade species *D. sechellia* and *D. mauritiana*, as well as *D. yakuba* as a close outgroup species. None of the novel hpRNAs identified in *D. simulans* have orthologs in *D. melanogaster* and *D. yakuba*, while most of them exist in the syntenic positions of the *D. sechellia* and/or *D. mauritiana* genomes (**Fig 2H**). These analyses clearly support that the vast majority of these hpRNAs were born in the *simulans*-clade ancestor, and two in *D. simulans* itself. Overall, the radical and asymmetric expansion of hpRNAs in the *simulans*-clade lineage, compared to *D. melanogaster*,

strongly suggests that the RNAi pathway has been deployed adaptively in these sister species, putatively to address emerging regulatory situations such as intragenomic conflicts.

## Target network of *D. simulans*-specific hpRNAs

We next investigated targets of novel hpRNAs annotated in *D. simulans*. Although many of these hpRNAs are shared in other *simulans*-clade species, we will subsequently refer to *D. simulans*-specific targets since we define these with respect to functional derepression in RNA-seq data (and not solely on the basis of hpRNA/target homology). In our previous work in *D. melanogaster*, we observed that hpRNAs typically exhibit substantial complementarity to one or a few target genes, ranging from an individual siRNA to extended regions that encompass multiple siRNAs [19, 26]. On this basis, we proposed that hpRNAs typically derive from their targets [19], analogous to plant miRNAs [27]. This set the stage that it seemed plausible, if not likely, that de novo hpRNAs in *D. simulans* might also have overt complementary targets. Indeed, we were able to identify compelling targets with antisense matching to most newly-recognized *D. simulans*-specific hpRNAs (**S1 Table**). In the following sections, we describe notable insights from specific aspects of the *D. simulans* hpRNA target network.

## Unexpected complexity in the *D. simulans* hpRNA network related to Dox family loci

Multiple X-linked Dox family genes, including two newly-recognized members (*PDox1* and *PDox2*), share an HMG-box domain that is derived from protamine and are targeted by newly-emerged hpRNAs (*Nmy*/*Tmy* class) [14–16]. Now, with expanded *D. simulans* hpRNA-target maps based on functional genomics, we reveal additional, *de novo* innovations within the Dox/hpRNA regulatory network.

We recently identified a sub-lineage of Dox-related loci that lack the HMG-box [15]. Via synteny comparisons with *D. melanogaster*, we inferred these to derive originally from fusion of a protamine-like copy in between *CG8664* and *forked* loci in an ancestor of *simulans* clade species, termed "original Dox" (*ODox*, **Fig 3A**). Our evolutionary tracing supports that *ODox* spawned the contemporary *Dox* family genes *PDox*, *UDox*, *MDox* and *Dox* across *simulans* clade Drosophilids [15]. Perhaps confusingly, then, the *ODox* locus in contemporary *simulans*-clade species retains segments of *CG8664* and the 5' UTR of *protamine*, but has lost its HMG-box (**Fig 3B**). *ODox* subsequently duplicated and mobilized to yield the related *ODox2* locus, which shares predicted domains with *ODox* and lacks an HMG-box, but also has divergent sequence material (**Fig 3B**). *ODox* and *ODox2* loci are proximal to centromere on the X chromosome at ~16Mb on the *D. simulans* long-read assembly [20], while the contemporary amplification of *Dox* family genes occurred at a distal genomic window of ~9-10Mb (**Fig 3B**).

As HMG-box domains seem relevant to the distorting functions of Dox family factors [15, 16], we wondered about impacts of Dox-superfamily loci lacking HMG-box domains. Unexpectedly, we realized that both *D. simulans ODox* and *ODox2* contain inverted repeats that bear functional hpRNA signatures, i.e., they generate Dcr-2-dependent small RNAs and their primary transcripts are upregulated in *dcr-2* mutants (**Figs 2, 3C, and 3D**). Accordingly, we renamed these loci *hp-ODox1* and *hp-ODox2*. Detailed analysis reveals further unanticipated features of their domain content. In particular, the inverted repeat at *ODox* bears ~200 bp homology to exon 2 of *Tapas*/*GD11509* and generates siRNAs with antisense complementarity to the parental *Tapas*/*GD11509* on chr2R (**Fig 3E**). In addition, *hp-ODox1* transcript contains a mix of repetitive sequences (**Fig 3C**).

Unexpectedly, the hairpin at *hp-ODox2* is not homologous to the hairpin in *hp-ODox1*, despite their shared lineage. Instead, the *hp-ODox2* inverted repeat contains sequence from

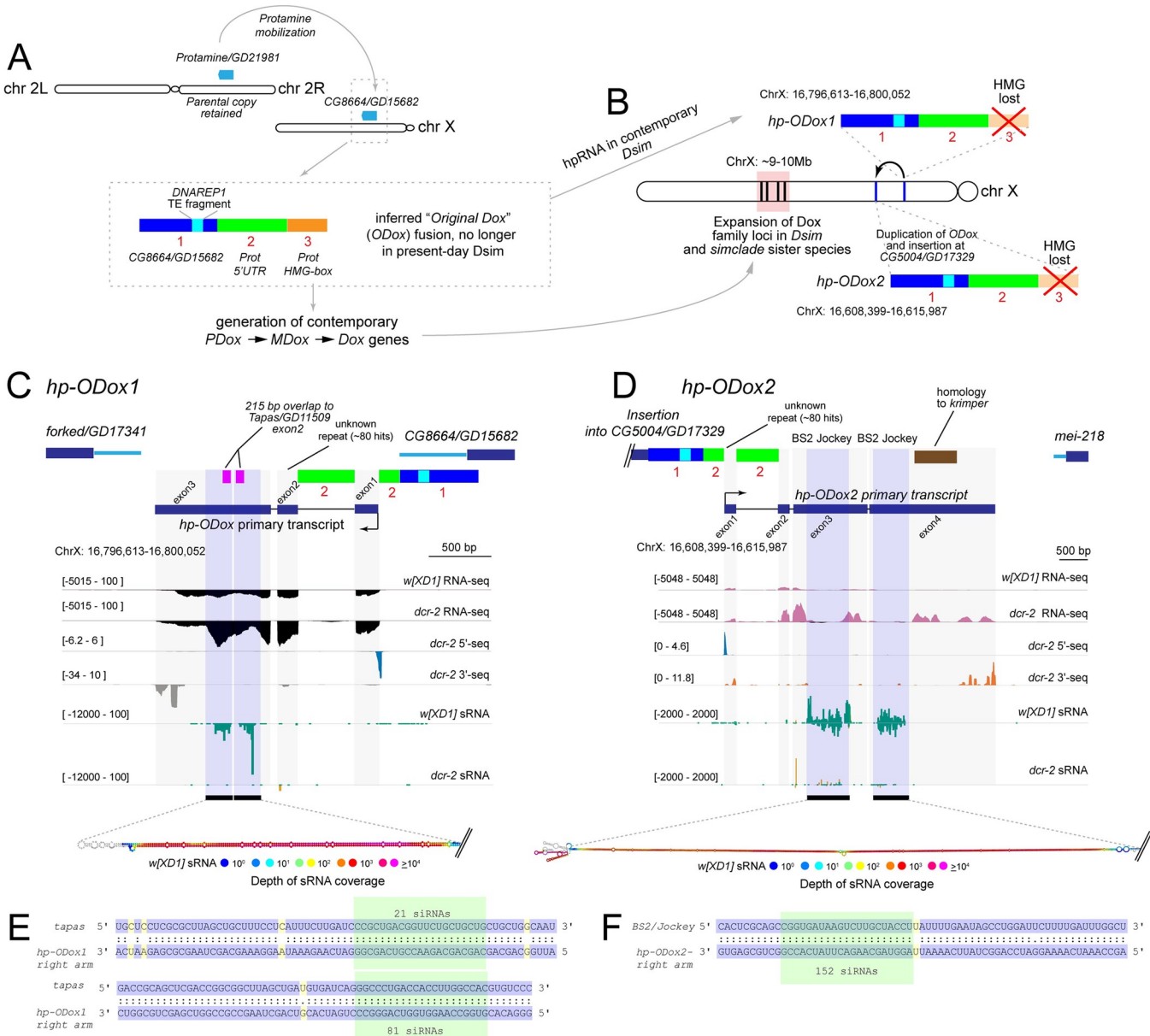

**Fig 3. Innovations in the Dox-related hpRNA network.** (A) Schematic of the evolution of *Dox* family genes, and the contemporary state of inferred, ancestral fusion of *Protamine* and *CG8664* (*Prot-CG8446* fusion) that birthed the *Dox* family genes. Segments 1, 2 and 3 in the inset box (inferred *ODox* fusion), shows genomic regions corresponding to their sequence of origin. Segment 1 (blue) is derived from *CG8664*, segment 2 (green) is derived from the 5' UTR of *Protamine*, and segment 3 (orange) contains *Protamine* coding sequence, including the HMG-box domain. Within segment 1, is a fragment of *DNAREP1* TE (turquoise). While contemporary Dox family genes *Dox*, *MDox*, and *PDox* share this segment structure, we infer that the ancestral "*ODox*" locus lost the HMG segment in contemporary *D. simulans* (B). In contemporary *D. simulans*, the ancestral *ODox* fusion is an hpRNA (*hp-ODox1*) and its duplication and insertion at *CG5004/GD17329* hosts another hpRNA (*hp-ODox2*). (C) *hp-ODox1* is a discrete hpRNA locus, revealed by upregulation of its primary hairpin transcript in *dcr-2* RNA-seq, presence of Dcr-2-dependent siRNAs, and demarcated by 5'-seq and 3'-seq. The inverted repeat arms of *hp-ODox1* bear homology to *Tapas* (pink), and other regions of *hp-ODox1* bear TE sequences. (D) *hp-ODox2* bears all the genomic signatures of an hpRNA, and its inverted repeat arms are homologous to *BS2/Jockey* TE. *pri-hp-ODox2* also bears homology to *Krimper*, although this sequence is not part of the inverted repeat and does not yield siRNAs. (E) Sequence alignment between *hp-ODox* and *tapas*. Examples of siRNAs with fully complementarity to *tapas* are boxed in green. (F) Antisense complementarity between *BS2/Jockey TE* and *hp-ODox2* with examplar siRNAs highlighted in green.

the *BS2/Jockey* transposable element (TE), generating siRNAs with antisense complementarity to *BS2/Jockey* (**Fig 3F**). In addition, other regions of the *pri-hp-ODox2* transcript bear other repetitive sequences as well as fragments from *Krimper*. However, the inverted repeat does not include *Krimper* sequence, and *Krimper*-targeting siRNAs were not observed.

Overall, we uncover additional innovations of the Dox-family lineage, which extend beyond HMG-box/protamine domains. In particular, derivatives of an ancestral "*ODox*" gene now generate hpRNA-siRNAs in contemporary *D. simulans*, and may be engaged in distinct genetic conflicts with connections to TE biology (**Fig 3**).

## Innovation of *D. simulans* hpRNAs that target other HMG-box loci

In addition to *Dox* family genes targeted by hpRNAs, we also find evidence that another class of HMG-box domain containing genes (testis HMG) is also targeted by novel hpRNAs in *D. simulans*. The Dox family genes belong to a distinct "MST-HMG-box" subclass [28], owing to their relatedness to *Drosophila* protamines [15, 16]. tHMG is not formally included in the MST-HMG-box family, but is nevertheless also testis-specific and expressed at highest levels during the histone-to-protamine transition [29]. We note evidence for rapid evolution of testis HMG-box loci, since protamine is locally duplicated in *D. melanogaster* (*MST35Ba/Bb*), but exists as a single copy in simulans clade species [15, 28]. Similarly, tHMG is locally duplicated in *D. melanogaster* (*tHMG1* and *tHMG2*), but bears a single copy in the syntenic region of *D. simulans* (**Fig 4A**).

Our functional profiling of *D. simulans* RNAi mutant testis allows us to discern further evolutionary dynamics of HMG-box-related hpRNAs (**Fig 4A**). First, we find that *D. simulans* contains a locus related to autosomal *D. melanogaster tHMG2*, which has mobilized to the X chromosome (**Fig 4A**). Detailed inspection shows that this locus is actually an hpRNA, as it generates Dcr-2-dependent small RNAs, and is associated with a spliced transcript that is derepressed in *dcr-2* mutants (**Fig 4B**). As is the case with certain other strong inverted repeat loci, the RNA-seq signal is poorly represented in the duplex arms of the hairpin, which resides near the 5' end of the primary transcript. However, the presence of capped and polyadenylated species provide experimental evidence for its termini. Of note, there is a local tandem duplication of the left arm of the hairpin, including within 5'-end mapping data (**Fig 4B**), and hints that its genesis as an hpRNA involved local duplications of a transposed sequence.

There are further complexities. We observe recent amplification of *tHMG2* in the heterochromatin boundary of chromosome X (Xhet), associated with numerous tandem hpRNAs within ~27kb (*hp-tHMG-cluster*) (**Fig 4C**). Based on the local duplication of *tHMG2*, and hpRNA secondary structure, we classify the 13 *hp-tHMG-cluster* hpRNAs into 3 subcategories (**S5 Fig**). The *hp-tHMG-cluster* also harbors three paralogs of *tHMG2* that are not part of inverted repeats. Of these, two paralogs appear to be full-length copies (88aa, compared to *D. simulans tHMG2* ortholog at 91aa), while a truncated paralog within this cluster encodes for only 24 aa (**Figs 4C and S5**).

At present, the nature of the primary transcript for the *hp-tHMG* cluster is uncertain, as near perfect complementarity of inverted repeats results in depletion of RNA-seq signal within *tHMG* hpRNA copies. We document 13 local inverted repeats (hpRNA copies) within the *hp-tHMG*-cluster. Of these, 6 have flanking 5'-end and 3'-seq signals, along with RNA-seq mapping between the hairpin arms that is upregulated in *dcr-2* mutants. While it is unclear how many individual transcription units exist with this hpRNA cluster, it is clear that the solo and clustered tHMG hairpin loci (located on opposite ends of the *D. simulans* X chromosome) are evolutionarily related, to each other and to the parental autosomal copy of tHMG (**Figs 4C and S5**). Indeed, we identify abundant siRNAs generated from *hp-tHMG-solo* and the *hp-*

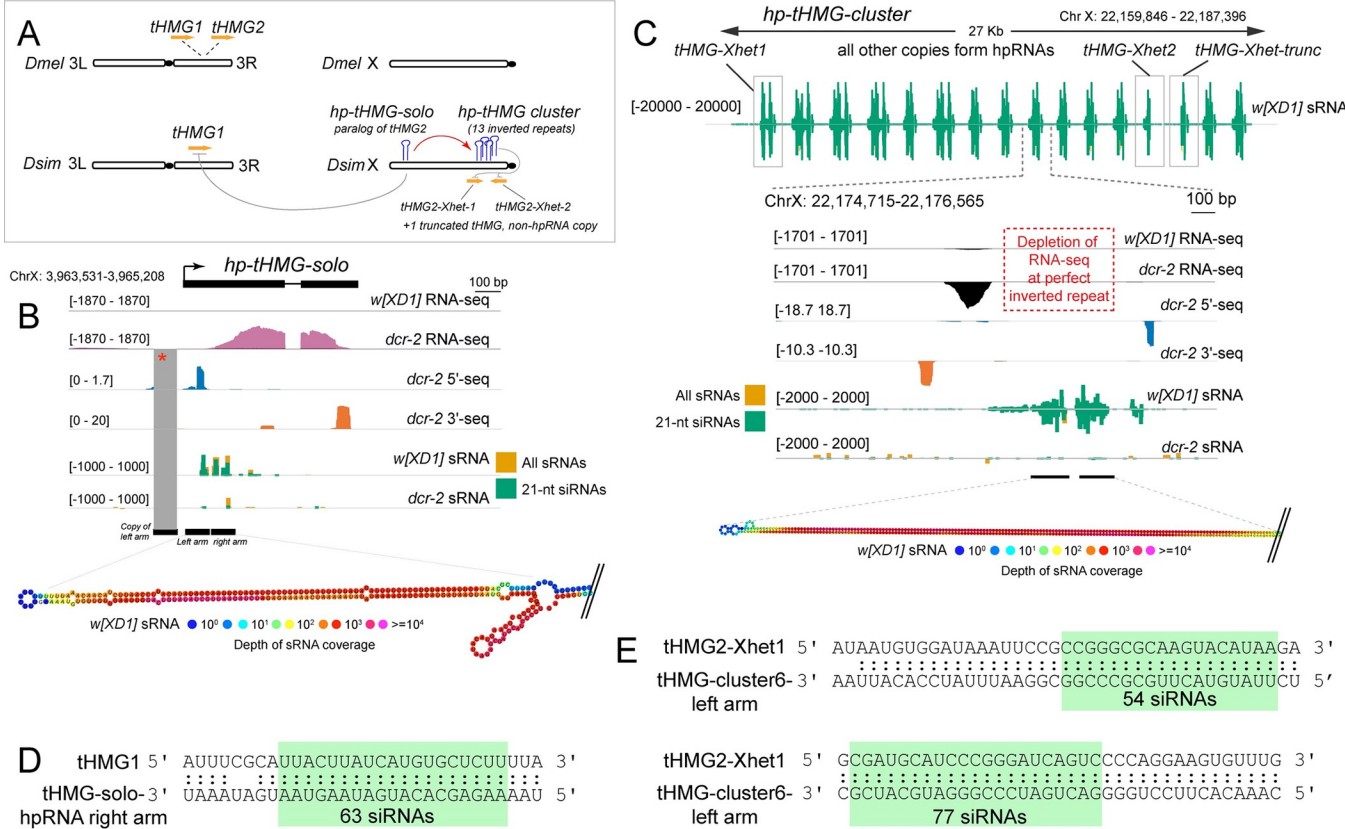

**Fig 4. Novel *D. simulans* hpRNAs related to testis HMG-box (tHMG) loci.** (A) Schematic of tHMG loci in *D. melanogaster* and *D. simulans*, and the origins of hpRNA suppressors and targets encompassing multiple novel X-linked tHMG-derived loci. tHMG is locally duplicated in *D. melanogaster*, but the syntenic location of *D. simulans* bears a single gene. However, *D. simulans* X chromosome bears multiple *de novo* hpRNAs with homology to tHMG, a single copy locus and another genomic cluster containing 13 tandem hpRNAs. (B) Genomic details of the *hp-tHMG-solo* locus, which is a spliced transcript with defined 5' and 3' ends, which generates abundant small RNAs from its duplex arm. Note that the 5' region of *hp-tHMG-solo* has been locally duplicated (red asterisk). (C) Genomic details of the *hp-tHMG-cluster* locus. This region bears 13 nearly identical hpRNAs, although the nature of the primary transcript(s) is not clearly evident, although 6 cluster units show evidence for 5'-seq and 3'-seq (detailed in **S6 Fig**). RNA-seq data across the tHMG-cluster locus appears depleted in the hpRNA duplex regions. Small RNA tracks show Dcr-2-dependent siRNAs from each cluster member. (D, E) Examples of antisense complementarity between tHMG genes and hp-tHMG-siRNAs, including autosomal *tHMG1* and *hp-tHMG-solo*, and *tHMG-Xhet1* and tHMG cluster loci.

*tHMG-cluster* that exhibit perfect complementarily to autosomal *tHMG1* and/or novel paralogs of *tHMG2* on Xhet (**Fig 4D**).

We analyzed the phylogenetic relationships amongst the general family of testis-restricted HMG-box factors in *D. melanogaster* and *D. simulans* (**S6 Fig**), which emphasizes that tHMG factors are an outgroup to the MST-HMG-box family [28], of which protamine/Dox family loci comprise a distinct subfamily. Interestingly, members of all these groups seem to be involved in sex chromosome conflicts [15–17, 30]. Altogether, these data emphasize greater co-evolutionary arms races between hpRNAs and their HMG-box targets than currently recognized.

## Molecular delineation of the stages of hpRNA emergence and evolution

With broader evidence that hpRNAs generally target specific genes, or groups of related loci, we sought broader perspective on evolution of hpRNA regulatory networks. Although all *D. melanogaster* hpRNAs emerged relatively recently [19], it is instructive to note that all *D. melanogaster* hpRNAs are conserved in *D. simulans* (**Fig 2E and 2F**). Therefore, we may consider

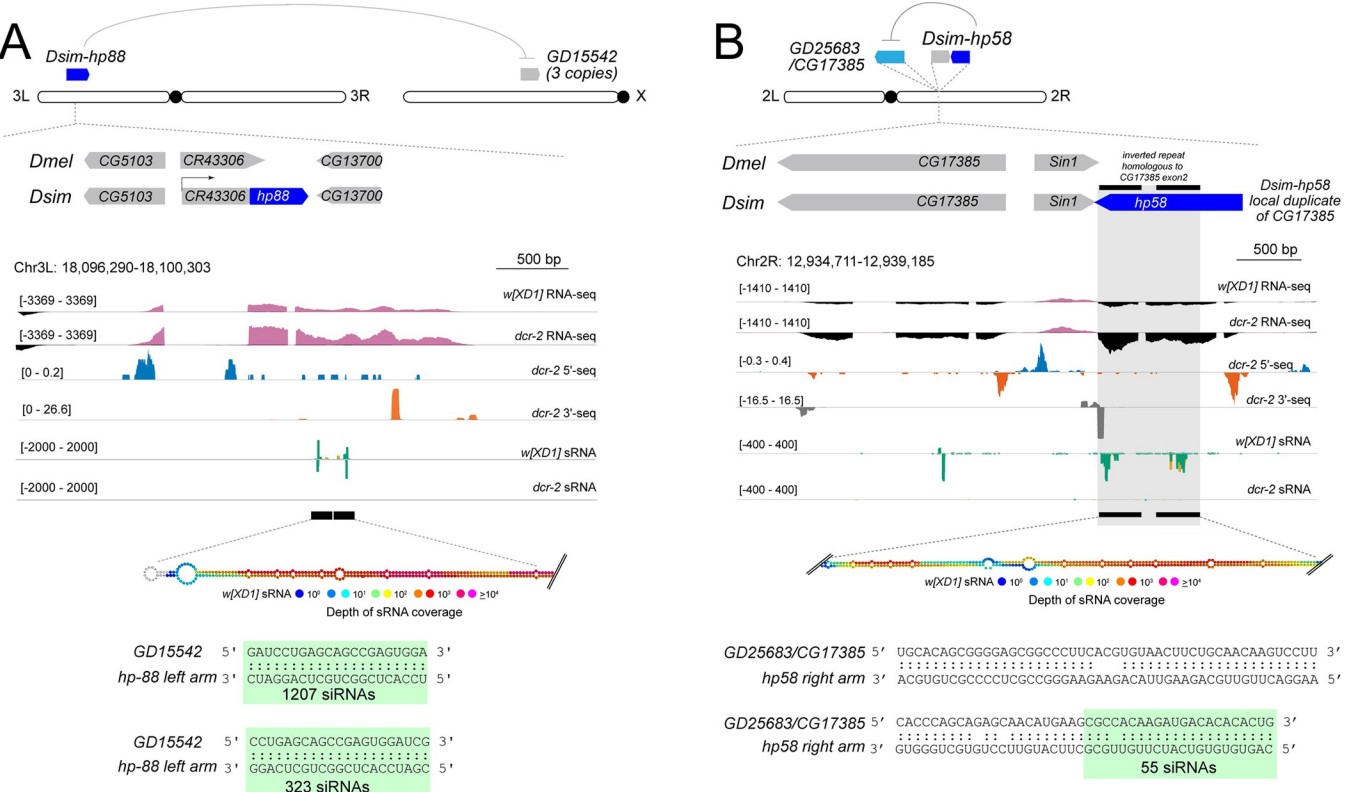

**Fig 5. Genomic features of novel *D. simulans* hpRNAs inform early stages in their birth.** (A) *Dsim*-hp88 emerged within the 3' UTR of non-coding RNA CR43306, which is syntenic between *D. melanogaster* and *D. simulans*. The inverted repeat arms bear homology to *de novo* paralogs on the X chromosome *GD15542* (3 copies expressed, but there are 13 other paralogous sequences of this novel gene on the X). *GD15542* is upregulated in *dcr-2* mutant indicating functional suppression via RNAi. hpRNA-target alignments for both *Dsim-hp58* and *Dsim-hp88* are provided with highlighted green box showing fully complementary siRNAs to their targets, *CG17385* and *GD15542*, respectively. (B) *Dsim-hp58* is a *de novo* hairpin that was born one gene away from its progenitor gene *CG17385*. It bears an inverted repeat fragment of exon 2 of *CG17385*. Similar to *pri-hp58*, *CG17385* is upregulated in *dcr-2* mutant testis RNA-seq data, indicating its functional suppression via the RNAi pathway.

*D. melanogaster* hpRNAs to be relatively older, compared to the numerous *de novo D. simulans* hpRNAs we identify in this study (**Fig 2C and 2G**). In principle, then, this collection of extremely young *D. simulans* hpRNAs may illuminate the earliest evolutionary stages of hpRNA birth.

An ongoing conundrum concerns "where" hpRNAs come from, especially as the functionally validated hpRNA-target relationships comprise examples where the hpRNA is genomically distant from its target [14, 19, 26, 31]. This could be due to mobilization of the hpRNA locus itself, or to derivation of the hpRNA via retrogene insertion [9, 11]. In the latter case, acquisition of a promoter may be an issue. We note that formation of *D. simulans hp88* occurred with the 3' region of an existing apparent non-coding locus *CR43306* (**Fig 5A**), suggesting that hpRNAs could take advantage of pre-existing transcription units for their expression.

Our catalog of *de novo* hpRNAs includes many other hpRNA loci that are genomically distant from their targets (**S1 Table**). However, the precedent of the interweaved locations of multiple related hpRNAs and targets of the *hp-pncr009/825-Oak* families in *D. melanogaster* suggested that some hpRNAs might be born from a genomic location close to their target. We now find several examples of this. For example, *D. simulans hp58* is a newly-emerged hpRNA that is adjacent to its pre-existing target gene *GD25683/CG17385* (**Fig 5B**). Similarly, we identify a novel hpRNA located at *D. simulans ballchen* (*GD18116*), created by a partial duplication

of the 5' region adjacent to the conserved *ballchen* gene annotation. This is reminiscent of the partial 5' duplication of the *hp-tHMG-solo* locus (**Fig 4**).

We also emphasize that hpRNA evolution frequently involves emergence of multiple copies. Beyond the described examples of expanding *hp-pncr009* family clustered loci and *hp-CG4068* tandem hpRNAs, we discovered novel amplified hpRNAs in *D. simulans*. These include tandem duplicates (as in the novel tHMG-related hpRNA cluster), local genomic duplicates of independent transcription units (as in de novo *hp-pncr009* family members, **S3B Fig**), or genomically dispersed copies (as in the case of four copies of the *hp70* family, **S1 Table**). Finally, we highlight *D. simulans hp88*, which was appended to an existing non-coding locus, but contains genetic material from the trio of *GD15542* gene copies located on the X chromosome (**Fig 5A**), much like amplifications of Dox family genes trigger hpRNA birth [15, 16].

## Functional repression by hpRNAs is most overt for the youngest siRNA loci

With an expanded view of hpRNAs and targets in hand, we addressed the larger consequences of RNAi loss on the testis transcriptomes of both species. Interestingly, although cytological consequences of RNAi loss on spermatogenesis is substantial in *D. melanogaster* [19] and severe in *D. simulans* [14], their respective transcriptome responses were relatively restricted (**Fig 2A and 2C**). For example, in *D. melanogaster*, only 245/16773 annotations detected >10RPM were 2-fold upregulated in *dcr-2* mutants compared to *dcr-2/+* heterozygous controls (FDR <1%) (**S2 Table**). However, all hpRNAs (except *hpRNA1*, upregulated only 1.5-fold) were amongst the top 20 upregulated transcripts in *dcr-2* mutants (**S2 Table**), confirming their efficient metabolism by Dcr-2. hpRNA targets also responded directionally to RNAi loss, in that none were downregulated. However, many hpRNA targets were not detected in adult testis. Of the 8 conserved hpRNA loci between *D. melanogaster* and *D. simulans*, the *pncr009* cluster hpRNAs (*hp-pncr009*, *hp-CR32205*, and *hp-CR32207*) bear homology to 10 target genes belonging to the *825-Oak* family [19] (**S3 Fig**). In *D. melanogaster dcr-2* mutant testis, all *825-Oak* family genes were <10RPM threshold, or indeed not detected (**S7 Fig**). The *D. melanogaster* orthologs of *825-Oak* loci are restricted to pupal gonads (http://flybase.org/), even though their corresponding hpRNA-siRNAs are detectable in adult testis. Of the remaining *D. melanogaster* hpRNAs targets, only two (*ATP-synthase ß* and *mus308*, targeted by *hpRNA1* and *hp-CG4068* respectively), were elevated in *dcr-2* mutants (**Fig 6A**). The targets of *D. melanogaster-D. simulans* conserved hpRNAs exhibited similar expression profiles in *D. simulans dcr-2* mutants (**Figs 6B and S7**). Thus, there is functional repression of targets of the older, conserved hpRNAs in both species, but the effects are generally modest. Nevertheless, the regulatory effects on siRNA targets are still greater than with most miRNA targets [32].

In striking contrast, the targets of the youngest siRNA loci, i.e. of *D. simulans*-specific hpRNAs, showed substantial greater directional change upon *dcr-2* loss (**Fig 6C**). In *D. simulans*, 1021/15119 loci expressed >10RPM exhibit at least two-fold upregulation in *dcr-2* mutant compared to *w[XD1]* (FDR <1%) (**S3 Table**). Of these, 22 novel and 7 conserved hpRNAs were in the top 200 derepressed genes. Moreover, 14/20 targets of novel hpRNAs in *D. simulans* were among top 200 upregulated genes (**S3 Table**). Of note, more genes were deregulated in *dcr-2* mutants in *D. simulans* compared to *D. melanogaster* (1021 vs. 245), consistent with the more severe cytological defects in *dcr-2* mutant testis of *D. simulans* (Lin et al. 2018) [14]compared to *D. melanogaster* {Wen et al and Lai Molecular Cell 2015}. To further assess whether the upregulation of *de novo* hpRNA targets is specific, we compared *de novo* hpRNA targets to related paralogs in *D. simulans* that are also expressed in testis. For example,

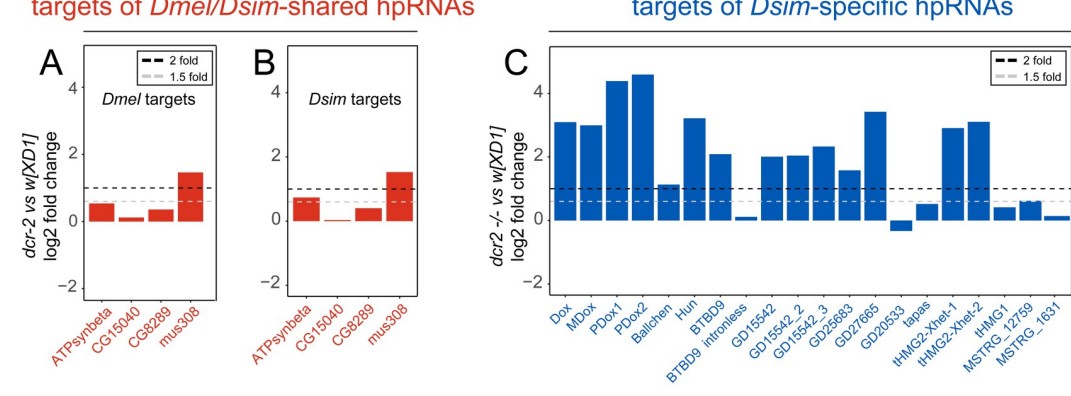

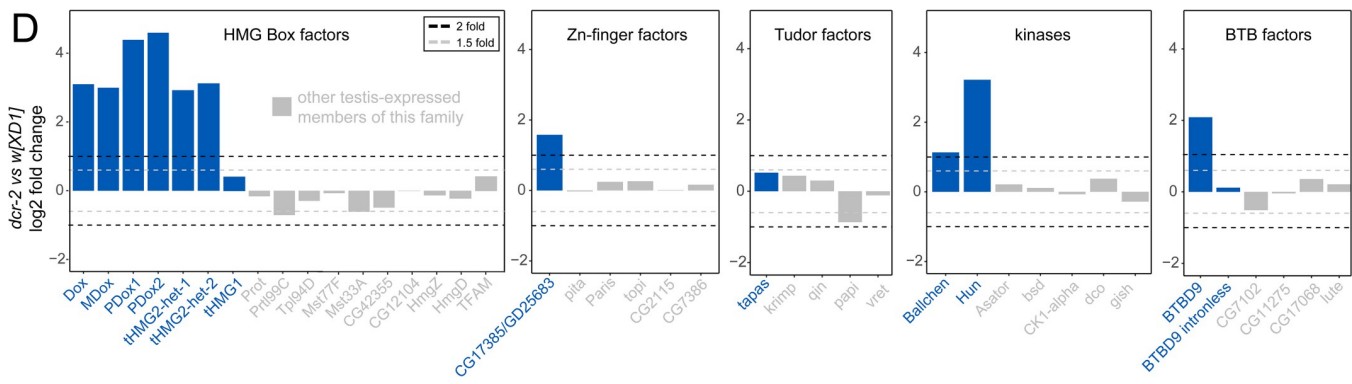

**Fig 6. Preferential suppression of targets of young hpRNAs vs. older hpRNAs.** Differential expression analysis of the targets of conserved and novel hpRNAs in *D. melanogaster* and *D. simulans* testis RNA-seq data. (A, B) Targets of hpRNAs conserved between *D. melanogaster* and *D. simulans*. Fold change values comparing wildtype and *dcr-2* mutant shown in yellow. Black and grey dotted lines show 2-fold and 1.5 fold changes in *dcr-2* mutant compared to wild type. Fold change values were estimated using two replicates each for the mutant and wildtype samples using DEseq package in R with an FDR < 1%. (C) Targets of *D. simulans*-specific hpRNAs (red) and their fold changes in *dcr-2* mutant compared to wild type. Note the functional repression of targets of young hpRNAs is much greater than for older hpRNAs. (D) The expression changes of additional testis-expressed genes with similar domains as the hpRNA targets (in grey) are plotted for comparison, which emphasizes specificity of upregulation of direct hpRNA targets (in red).

we detected directional change in expression of Dox family members upon loss of Dcr-2, but not their related HMG-box paralogs that are also expressed in testis (**Fig 6D**). Similar specificity was observed for other families of *de novo* targets (zinc-finger factors, Tudor proteins, kinases, and BTBD factors, **Fig 6D**). However, we wish to note that the mRNA for the tudor protein Tapas is not selectively upregulated, although *hp-ODox*-1 generates a population of siRNAs antisense to *Tapas*. In accord with the striking derepression of the targets of *de novo* hpRNAs in RNA-seq data, they are targeted by far more antisense siRNAs than are targets of hpRNAs that are conserved between *D. simulans* and *D. melanogaster* (**S4 Table**).

Overall, the analysis of *D. simulans* was particularly informative, since we could compare the properties of "young" and "older" hpRNA-target interactions. This was perhaps unexpected, since we had earlier used the latter cohort to derive clear evidence of adaptive co-evolution between hpRNAs and their targets [19]. Due to insufficient orthologs, we lack statistical foundation to assess co-evolution with *simulans*-specific hpRNAs. Nevertheless, the picture is clear that younger hpRNAs mediate quantitatively greater target suppression than older hpRNAs. In particular, these data flip the rationale for small RNA mediated regulation relative to the miRNA pathway, for which recently-evolved loci might generally be neutral and only the oldest miRNAs appear to have biologically significant effects [33].

## Biased X-linkage of *de novo D. simulans* hpRNA targets, along with some hpRNAs, reflects likely roles in sex chromosome conflict

*D. simulans* hpRNAs that are conserved in *D. melanogaster* preserve their documented targeting of a cohort both young and ancient genes [19]. Overall, the targets of these conserved hpRNAs are not biased in their genomic location, as they are distributed across all the chromosomes (with multiple targets on each of the larger chromosomes, **S8 Fig**). Thus, there is no overt bias to the age and location of "old" hpRNA targets, beyond the fact that many hpRNAs and targets in the *hp-pncr009/825-Oak* network are clustered within a small genomic interval, and continue to expand actively (**Figs 7** and **S3**).

On the other hand, a distinct pattern emerges with our collection of "young" *D. simulans* hpRNAs. These very young hpRNAs (blue loci, **Fig 7A**) are distributed across the major chromosomes, in a pattern relatively similar to the older hpRNAs (orange loci, **Fig 7A**). On the other hand, the targets of these recently-emerged *D. simulans* hpRNAs, exhibit statistically significant bias for X-chromosome localization (Fisher's exact test, $p = 0.0416$; blue loci, **Fig 7B**). In addition to Dox and MDox, 14/20 (70%) other targets of *D. simulans*-emerged hpRNAs are found on the X, whereas only 2/11 (18%) targets of hpRNAs shared with *D. melanogaster* are found on the X. These genes are all testis-specific paralogs of older gene families, and many encode proteins with roles putatively related to meiosis or chromosome segregation. As documented, these include four paralogous loci Dox, MDox, PDox1 and PDox2, which define a rapidly evolving set of young protamine-like gene copies with known or inferred meiotic drive activities [9, 11, 14–16]. Moreover, we reported the existence of additional X-linked tHMG loci, for which several protein-coding copies within the tHMG-cluster and the Xhet region are targeted by hpRNAs (**Figs 4, S5, and S6**).

*Hun Hunaphu* (*Hun*) is another intriguing X-linked hpRNA target. *Hun* is a newly-emerged X-linked derivative of *ballchen*, encoding a histone kinase that is essential for

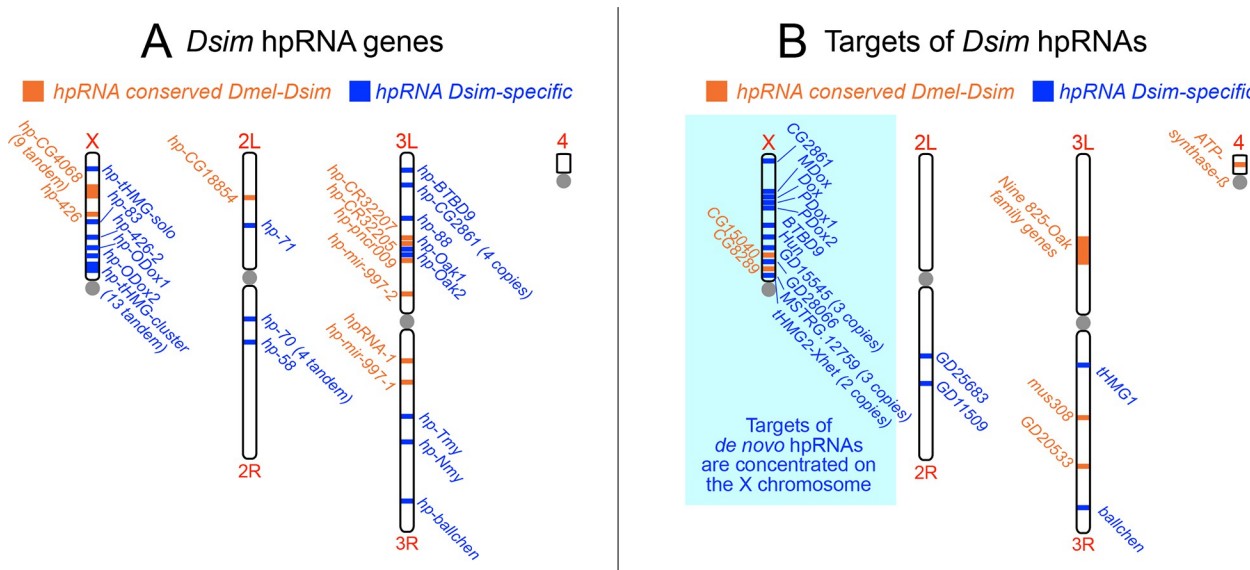

**Fig 7. Biased location of *de novo* hpRNA targets on the *D. simulans* X chromosome.** (A) Genomic location of *D. simulans* hpRNAs, separated into ones that are conserved in *D. melanogaster* (in orange) and ones that are *D. simulans*-specific (in blue). Both classes of hpRNAs are distributed across all the major chromosome arms. (B) Genomic location of *D. simulans* hpRNA targets, separated into ones whose hpRNA is conserved in *D. melanogaster* (in orange) and ones whose cognate hpRNA is *D. simulans*-specific (in blue). The target of *hp-ODox-2* (*BS2/Jockey*) was not included in this visualization, as the *BS2/Jockey* transposable element has numerous copies across all major chromosomes. The concentration of targets of *D. simulans*-specific hpRNAs on the X suggests that they may comprise novel selfish genes involved in *sex ratio* meiotic drive.

germline stem cell renewal (in both sexes) and meiotic chromosomal architecture [34–37]. However, like Dox family genes [15, 16], *Hun* is a chimeric, young gene that has lost parts of Ballchen and gained new coding sequence [38]. Of note, Hun orthologs exhibit a large excess of nonsynonymous substitutions compared to Ballchen [38]. Ballchen is moderately upregulated in *dcr-2* mutant testis but *Hun* is much more strongly derepressed (**Fig 6**). In light of their strong suppression by endo-RNAi, we take this as a strong suggestion for a selfish function of Hun that necessitates silencing by a hpRNA. Overall, we infer the biased location of target genes of *de novo* hpRNAs on the X chromosome reflects its disproportionate involvement in active genomic conflicts.

## Discussion

### The hpRNA targeting landscape is diametrically opposite to miRNA regulation

We conceive two general classes of hpRNA targets. While hpRNAs are all evolutionary young, some of their targets are relatively old genes; e.g. targeting of *ATP synthase-ß* by hpRNA-1. We imagine there are adaptive reasons for why mild suppression of such loci imparts beneficial regulatory consequences. This may be due directly to the acquisition of elevated transcriptional properties of the targets, and/or by emergence of duplicated loci with preferred testis expression, which appears to be a relatively common process [39, 40]. In any case, the role for endo-RNAi here is to modulate target activity, since full suppression of these genes is clearly deleterious. Still, we may speculate further that as several well-conserved targets of hpRNAs encode protein activities that have been linked to speciation, such as heterochromatin, DNA damage, and energy homeostasis [19], these conserved genes may harbor selfish activities in certain species that warrants adaptive suppression by RNAi. Since the miRNA pathway is not typically involved in adaptive targeting, and instead relies upon capture of targets bearing invariant miRNA seed matches, the RNAi pathway may be more flexible to suppress such genes. A counterpoint to this is the fact that specific testis-expressed, evolutionarily young, miRNA clusters diverge even within their seed regions [21]. This suggests a possible atypical role for specialized miRNA loci in "hpRNA"-like evolutionary targeting dynamics.

### Recurrent targeting of HMG-box factors in distinct hpRNA/RNAi networks

Our annotation of hpRNAs in two species indicates that, as a rule, these endogenous siRNA loci comprise very short-lived genes. Thus, they can at best only mediate modestly conserved regulation. If this is the case, can we learn any general principles from such fast-evolving regulatory networks? In fact, when considering this study alongside recent literature, we find several recurrent themes that provide a framework for understanding how endogenous RNAi is harnessed in biology.

We recently found that *de novo* hpRNAs in the *simulans* clade are required to silence a newly-emerged, amplifying, and selfish set of X-linked protamine derivatives, namely the Dox family [15, 16]. Protamines are central factors that condense the sperm genome, and therefore seem ripe for co-option by selfish factors to disrupt paternal inheritance. Indeed, following removal of histones, multiple sperm nuclear basic proteins (SNBPs) play roles in packaging sperm chromatin, and most of these contain HMG-box domains. We find that beyond the X-linked Dox family, there is a separate amplification of X-linked tHMG box genes in *D. simulans* that are concomitantly associated with silencing by cognate hpRNAs. Thus, we infer there is recurrent innovation of selfish SNBP activities by X chromosomes, consistent with the

notion of sex chromosome meiotic drive that requires silencing by endogenous RNAi. Prot-amines are also functionally relevant to activity of *Segregation Distorter*, an autosomal meiotic drive system in *D. melanogaster* [41, 42]. Moreover, the Malik group recently reported high turnover of testis HMG-box loci across the Drosophilid genus, supporting the notion that their rapid evolution is due to recurrent intragenomic conflict between sex chromosomes {Chang and Malik eLIFE 2023}. We predict that hpRNAs may be employed to silence other protamine-based meiotic drive phenomenon in other species.

Within the Dox superfamily system itself, we document the innovation of novel hpRNAs that bear chimeric domain structures characteristic of coding Dox genes, but that lack the HMG-box. Instead, they incorporate sequences from the piwi-interacting RNA (piRNA) factors Tapas and Krimper [43, 44], also bear TE fragments. Homology to piRNA factors Tapas and Krimper were recognized in a prior study in distinct duplicate copies of the ancestral "ODox" gene, termed X:17.1 and X:17.2 [16]. However, our analyses revealed that the duplicate copies of the ancestral *ODox* gene bear inverted repeats and now generate hpRNA-siRNAs (two independent loci, which we termed *hp-ODox1* and *hp-ODox2*) in contemporary *D. simulans* (**Fig 3**). Overall, the genetic conflict that led to identification of *Dox* and *Nmy* [9, 11] is actually part of a far more extensive and rapidly evolving network of putative meiotic drivers and suppressors, and may in fact integrate activities of the siRNA and piRNA pathways. Relevant to this, the third recognized *sex ratio* meiotic drive system in *D. simulans* ("Paris"), is driven by HP1D2 [13], a derivative of the core piRNA factor Rhino [45, 46]. Thus, there are recurrent linkages of piRNA factors to *sex ratio* meiotic drive, notwithstanding that TEs are themselves selfish genetic elements.

Of course, TEs are intrinsically selfish elements that are targeted by host genomic defenses, most famously by piRNAs. However, recent studies provide analogous conceptual involvement for co-option of the piRNA pathway by drive systems. As mentioned, the Paris SR system utilizes a *de novo* copy of the HP1-like factor Rhino [13], a central nuclear piRNA factor that defines piRNA cluster transcription [45, 46]. As another example, telomeric TART elements were found to have captured a fragment of *Nxf2* [47], a piRNA-specific copy of the mRNA export machinery that gained activity in co-transcriptional silencing [48–51]. We surmise that the capture of some piRNA factors by hpRNA loci may in fact reflect their selfish activities of such defense factors.

The adaptive deployment of hpRNAs in the testis in *D. simulans* highlights that some of the most important biologically overt manifestations of endo-RNAi cannot be studied in the major model system *D. melanogaster*. Looking to other Drosophilids, the recognition of rampant duplications of the RNAi effector AGO2 in various *obscura* clade species, resulting primarily in testis-restricted paralogs [52–55], provides a further hint into active genomic conflicts that may be playing out in these species. Indeed, intragenomic conflicts that mediate aberrant *sex ratio* and/or sterility, specifically in male fathers, have been documented in the *obscura* clade [56, 57]. The genetic factors in these conflicts remain to be documented fully, but it is intriguing to hypothesize whether *de novo* hpRNAs might be involved in any of these scenarios. Overall, the *Drosophila* RNAi/hpRNA pathway provides a policing system that helps to surveil and silence gene expression in the testis against selfish meiotic loci. We speculate that such cycles of drive and repression are poised to underlie speciation [5, 58, 59].

## Materials and methods

### Drosophila strains

Stocks bearing *D. melanogaster dcr-2* alleles *wIR; dcr-2[L811fsx]/CyO* and *wIR; dcr-2[R416X]/CyO* were obtained from Richard Carthew (Northwestern) [60]. *D. simulans w[XD1]* wild-

type strain was obtained from BestGene, Inc. and used as the control strain for mutant comparisons. *dcr-2* loss of function mutant with *DsRed* allele replacing the endogenous locus was made in the *w[XD1]* strain background as reported in [14]. Similar to *dcr-2* DsRed mutant allele, we also generated *dcr-2* mutant allele replacing the endogenous locus with a *mini-white*[+] marker for efficient selection of *dcr-2* trans-heterozygous mutants by crossing *dcr-2[DsRed]* and *dcr-2[white*[+]*]* parental files. Oligo sequences for *dcr-2* targeting are listed in **S5 Table**. All flies were reared on standard cornmeal molasses food. As *D. simulans* lack balancer chromosomes, and as *dcr-2* homozygous mutants are also male sterile [14], we maintained *dcr-2 [DsRed]* and *dcr-2[white*[+]*]* alleles by visual selection of markers every few generations to maintain the alleles.

## Testis dissection and RNA preparation

For *D. melanogaster* testis dissections *dcr-2* mutant testis were collected from *wIR; dcr-2 [L811fsx]/dcr-2[R416X]* trans-heterozygotes and *wIR; dcr-2[R416X]/+* heterozygous flies were used as controls. For *D. simulans*, we collected testis from *dcr-2[DsRed]*/[white*[+]*] trans-heterozygous mutants, and used the parental strain *w[XD1]* as control. Briefly, testis from 3 days old flies were extracted in TRIzol (Invitrogen) in batches of 10 flies at a time and the testis samples were flash frozen in liquid nitrogen. RNA was extracted from 25–50 testis per genotype.

## Small RNA and RNA-seq library preparation

RNA extraction was performed as described in [14], and the quality of RNA samples were assessed with the Agilent Bioanalyzer. RNA samples with RIN >6.5 were used for library preparation using the Illumina TruSeq Total RNA library Prep Kit LT. Briefly, for RNA-seq libraries we used 650 ng of total RNA, and we used the Manufacturer's protocol except for reducing the number of PCR cycles from 15 as recommended to 8, to minimize artifacts that may arise from PCR amplification. We prepared stranded RNA-seq libraries for *D. simulans* and unstranded libraries for *D. melanogaster* as RNA samples were extracted and processed in different time points. Samples were pooled using barcoded adapters provided by the manufacturer and the paired-end sequencing was performed at New York Genome Center using PE75 in the Illumina HiSeq2500 sequencer.

We prepared small RNA libraries using ~20 μg total RNA, as previously described [14]. To the total RNA pool, we added a set of 52 RNA spike-ins, spanning a range of concentrations (QIAseq miRNA Library Spike-In kit #800100). A list of spike-in sequences used for small RNA library preparation is provided in **S5 Table**. Briefly, small RNAs of size 18- to 29-nt-long small RNAs were purified by preparative PAGE. Next, the 3′ linker (containing four random nucleotides) was ligated overnight using T4 RNA ligase 2, truncated K227Q (NEB), after which the products were recovered by a second PAGE purification. 5′ RNA linkers with four terminal random nucleotides were then ligated to the small RNAs using T4 RNA ligase (NEB) followed by another round of PAGE purification. The cloned small RNAs were then reverse transcribed, PCR amplified and sequenced using P50 single-end sequencing on the Illumina HiSeq 2500 sequencer.

## 5'-seq and 3'-seq library preparation

To map 5' ends, we used the parallel analysis of RNA 5′ ends from low-input RNA (nano-PARE) strategy [24]. For *D. simulans* libraries, testis was extracted from <1-week males and total RNA was extracted using TRIzol. cDNA was prepared using Smart-seq2 [61] and tagmented using the Illumina Nextera DNA library preparation kit, purified using the Zymo 5x DNA Clean and Concentrator kit (Zymo Research), and eluted with resuspension buffer. For

5'-end enrichment PCR, the purified reaction was split and amplified either Tn5.1/TSO or Tn5.2/TSO enrichment oligonucleotide primer sets. PCR reaction products with Tn5.1/TSO enrichment oligonucleotide and Tn5.2/TSO enrichment oligonucleotide primer sets were pooled and purified using AMPureXP DNA beads. Final libraries were checked for quality on an Agilent DNA HS Bioanalyzer chip. Libraries with size ranges between 150 and 800 bp were diluted and sequenced to 10–15 million single-end 50-bp reads per sample using a custom sequencing primer (TSO_Seq) and a custom P5/P7 index primer mix on an Illumina HiSeq 2500 instrument.

To annotate 3' transcript termini, we used the QuantSeq 3' mRNA-seq library preparation REV kit for Illumina (Lexogen) with a starting material of 50 ng total RNA from *D. melanogaster* and *D. simulans* control and *dcr-2* mutant samples, according to manufacturer's instructions. cDNA libraries were sequenced on Illumina HiSeq-1000 sequencer with single-end SE 50 mode.

### Genomic analyses of RNA-seq data in *D. melanogaster* and *D. simulans*

**RNA sequencing analysis.** Paired-end RNA-seq reads from wild-type and mutant *dcr-2* samples in *D. melanogaster* and *D. simulans* were mapped to dm6 (FlyBase) and *D. simulans* PacBio assemblies [20], respectively using hisat2 aligner [62, 63]. The resulting alignments in SAM format was converted to BAM using SAMtools software [64] for downstream analyses. Mapping quality and statistics were determined using the *bam_stat.py* script provided in the RSeQC software [65]. Transcript abundance was determined using FeatureCounts software from the subread package [66], using *D. melanogaster* gene annotations from FlyBase r6.25. For *D. simulans*, we used both gene annotations from FlyBase and *de novo* transcript annotation using StringTie software (see details below) [67]. As FlyBase gene annotations for *D. simulans* correspond to *D. simulans* r2.02 assembly, we converted the FlyBase assembly annotations to *D. simulans* PacBio coordinates using the UCSC liftover tool implemented in the KentUtils toolkit from UCSC (https://github.com/ENCODE-DCC/kentUtils). We combined FlyBase liftover and *de novo* annotations in *D. simulans* to determine transcript abundance for RNA-seq analyses. The following description for differential gene expression (DFE) analysis is the same for *D. melanogaster* and *D. simulans* data. DFE comparing control and *dcr-2* mutant data was performed using the DEseq2 package in R [68]. For visualization of mapped reads, the BAM alignment files were converted to bigwig format using *bam2wig.py* script from RSeQC [65] and the bigwig tracks were visualized on the IGV genome browser [69].

**Small RNA sequencing analysis.** Adapters were trimmed from small RNA sequences using Cutadapt software (https://github.com/marcelm/cutadapt); then the 5' and 3' 4-nt linkers (total 8 bp) were removed using sRNA_linker_removal.sh script described in [15] (https://github.com/Lai-Lab-Sloan-Kettering/Dox_evolution). The adapter and linker removed sequences were then filtered to remove < 15 nt reads. We mapped > 15 nt reads from *D. melanogaster* and *D. simulans* genotypes to dm6 reference genome assembly and *D. simulans* PacBio assembly, respectively, with Bowtie [70] using the following mapping options: bowtie -q -p 4 -v 3 -k 20—best–strata. The resulting BAM alignments from bowtie mapping were converted to bigwig for visualization using *bam2wig.py* script from the RSeQC software [65]. During the BAM to bigwig conversion step, the small RNA mapping data was normalized to 52 spike-in sequences from the library (QIAseq miRNA Library Spike-In kit). The normalization was performed across the 52 sequences to get a single value as indicated in the manufacturer's protocol as follows to obtain TPM for spike-in reads. TPM = (# of spike-in reads/total reads)*10[6]. In addition, according to manufacturer's protocol, after this simple normalization, we did correlation matrices for sample-to-sample comparisons. As indicated in the normalization

procedure, we observed a good sample-to-sample correlation for spike-ins ($R^2$ ranging from 0.95–0.99). The spike-in reads normalized to a million in the small RNA library was then used to normalize small RNA reads mapped to the loci of interest.

### *De novo* annotation of testis transcriptome

In addition to previously annotated transcripts/genes from the FlyBase annotation, we performed *de novo* annotation of our transcriptome data to identify additional, novel testis-expressed transcripts in *D. melanogaster* and *D. simulans*. The novel annotated transcripts were then supplemented with known annotations to make a combined set of 17285 transcripts in *D. melanogaster* and 15119 transcripts in *D. simulans*. We employed two independent, genome assembly guided transcript prediction algorithms, Cufflinks [71] and StringTie [67]. For both methods, *de novo* transcripts were predicted for each RNA-seq dataset, and a merged transcript model was generated encompassing the transcriptome from WT and mutant datasets. hpRNAs were predicted using the scheme shown in S2 Fig, and visualized using the Integrated Genomics Viewer (IGV) [72]. The termini of primary hpRNA transcripts were refined using the 5'-seq and 3'-seq data.

### Homology searches for hpRNA orthologs

For each novel hpRNA identified in *D. simulans*, we first determined its genomic location based on flanking genes, and searched for homologous sequences (with synteny) in other closely related *simulans* clade species (*D. mauritiana* and *D. sechellia*), as well as *D. melanogaster* and *D. yakuba* as a close outgroup species. No matches to newly-identified *D. simulans* hpRNAs were recorded in *D. melanogaster* and *D. yakuba*, whereas most of these had hits in *D. mauritiana* and/or *D. sechellia*. In the latter cases, we assessed them further for presence at the syntenic location of the *D. simulans* hpRNA. This proved to be the case for all loci with homologs, in which case we infer their birth within the *simulans* clade ancestor. In some cases no hits were identified in other species, suggesting its potential birth in the lineage leading to *D. simulans*.

## Supporting information

**S1 Fig. Rationale and strategy to annotate hpRNAs.** (Top) Expectations of reciprocal behavior of hpRNA transcripts in wildtype and *dcr-2* mutants, with respect to RNA-seq and small RNA data. (Bottom) Overall strategy for identification of *D. melanogaster* and *D. simulans* hpRNAs. The overall procedures are similar, except that *de novo* transcriptome was generated for *D. simulans*, owing to its less well-annotated genome.
(TIF)

**S2 Fig. Additional examples of *D. melanogaster*-*D. simulans* conserved hpRNAs.** Shown are genome browser tracks that illustrate the reciprocal behavior of hpRNAs in control vs. *dcr-2* RNA-seq and small RNA data; all of these loci are shared in the syntenic locations between *D. melanogaster* and *D. simulans*. In all cases, *dcr-2* mutants stabilize a primary hpRNA transcript while losing the ~21-nt small RNAs from the duplex regions of the hpRNA.
(TIF)

**S3 Fig. Copy number changes in hpRNA clusters in *D. melanogaster* and *D. simulans*.** (A) Synteny alignment of pncr009 hpRNA cluster region on chr3L in *D. melanogaster* and *D. simulans*. In *D. melanogaster*, there are three hpRNAs in this region (*hp-CR32205*, *hp-CR32207*, and *hp-pncr009*, shown in red). The targets of pncr009 hpRNAs are in the vicinity of hpRNAs, and are shown in blue. Unrelated genes in the 92-kb window are shown in grey.

Synteny representation is shown as local colinear blocks of sequences derived from Mauve alignment using the Geneious software. Non-colinear regions in the syntenic alignment (white regions) are insertions in respective species. Two novel hpRNA paralogs in the *D. simulans* pncr009 region (*hp-Oak1* and *hp-Oak2*) are shown in green. (B) IGV screenshot of hp-CG4068 cluster in *D. melanogaster* and *D. simulans*. In *D. melanogaster*, there are 20 tandem copies of the hpRNA, while in *D. simulans* there are only 9 tandem copies. Small RNA tracks show loss of siRNAs in *dcr-2* mutant testes in both *D. melanogaster* and *D. simulans*. Expression of flanking genes is shown in the RNA-seq tracks.
(TIF)

**S4 Fig. Depletion of RNA-seq data within highly duplexed regions of primary hpRNA transcripts.** (A) *hp-Nmy*. The RNA-seq tracks show upregulation of pri-hpRNA in *dcr-2* mutant testis. However, the RNA-seq signal is not uniform across the primary transcript. Depletion of RNA-seq is evident within the red dotted box, corresponding to inverted repeat arms of the hpRNA. Small RNA tracks show that Dcr-2-dependent siRNAs are generated from the duplex region. (B) *hp-71* shows similar depletion of RNA-seq in the inverted repeat arm region (red dotted box).
(TIF)

**S5 Fig. Amplification and sequence arrangement of tHMG copies in the tHMG cluster region.** (A) 16 units within the X-linked tHMG cluster region are evident from small RNA mapping. Boxed regions show individual units and the tHMG-Xhet copies within each unit is shown in red and blue based on the duplicate copy orientation. Copies with an asterisk indicate partial/truncated copies of tHMG-Xhet. Within this cluster, tHMG-Xhet amplifications have birthed 13 hpRNAs and shown in green are hpRNAs for which there is absence of 5'-seq and 3'-seq evidence and 6 hpRNAs (blue box) shown in grey have 5'-seq and 3'-seq data. Shown in blue text are two copies of tHMG-Xhet (1 and 2) that are not part of an hpRNA arrangement. (B) Example tHMG arrangement of individual hpRNA unit. Shown in blue and red are orientation of paralog arrangement. Alignment of tHMG-Xhet paralogs with respect to tHMG-solo-locus is shown below. Note there is also small RNA mapping to the partial/truncated copy of the tHMG-Xhet paralog. Shown in black box on the alignment is a sequence window which is spliced at the tHMG-solo-locus.
(TIF)

**S6 Fig. Phylogeny of testis HMG-box domain containing factors in *D. melanogaster* and *D. simulans*.** HMG-box domain containing loci with testis-specific expression was identified in [73]. We compared the relationships of all testis-HMG-box domain containing genes, using testis HMG-box domain (tHMG) proteins as an outgroup. tHMG paralogs in *D. simulans* are targeted by hpRNAs (main Fig 4), while the Dox family genes that emerged from Protamine like ancestor (ODox, red rectangle) are targeted by hp-Nmy and hp-Tmy. In contemporary *D. simulans*, *ODox* and its duplicate are distinct hpRNAs (main Fig 4).
(TIF)

**S7 Fig. Expression of targets of *D. melanogaster*-*D. simulans* conserved hpRNAs and targets of novel *D. simulans*-specific hpRNAs.** Amongst conserved targets, we note that all members of the 825-Oak family show low to no expression in adult testis. Data from biologically independent RNA-seq experiments are shown as individual dots (wildtype in red and *dcr-2* mutant in black). Compared to targets of conserved hpRNAs, targets of *D. simulans*-specific hpRNAs show much greater derepression in *dcr-2* mutant testis.
(TIF)

**S8 Fig. Chromosomal maps of *D. melanogaster* hpRNAs and targets.** (Left) Locations of *D. melanogaster* hpRNAs. (Right) Locations of *D. melanogaster* hpRNA targets. Note that all *D. melanogaster*-specific hpRNAs have homologs in *D. simulans*, whereas there are numerous *D. simulans*-specific hpRNAs that are lacking in *D. melanogaster* (see main Figs 2 and 6).
(TIF)

**S1 Table. Annotation of conserved and novel hpRNAs in *D. melanogaster* and *D. simulans*.**
(XLSX)

**S2 Table. Upregulated genes in *D. melanogaster* *dcr-2* mutant testis.**
(XLSX)

**S3 Table. Upregulated genes in *D. simulans* *dcr-2* mutant testis.**
(XLSX)

**S4 Table. Quantification of hpRNA-derived antisense siRNAs mapped to hpRNA targets.**
(XLSX)

**S5 Table. Oligonucleotide sequences used in this study.**
(XLSX)

## Acknowledgments

We are grateful to members of the *simulans* clade PacBio sequencing consortium (J. J. Emerson, Amanda Larracuente, Colin Meiklejohn and Kristi Montooth) for access to *D. simulans* PacBio data at the unpublished stage. We thank Richard Carthew (Northwestern University) and the San Diego Drosophila Stock Center for fly stocks.

## Author Contributions

**Conceptualization:** Jeffrey Vedanayagam, Eric C. Lai.

**Data curation:** Jeffrey Vedanayagam, Ching-Jung Lin, Alex S. Flynt, Jiayu Wen.

**Formal analysis:** Jeffrey Vedanayagam, Ranjith Papareddy, Alex S. Flynt, Jiayu Wen, Eric C. Lai.

**Funding acquisition:** Jeffrey Vedanayagam, Eric C. Lai.

**Investigation:** Jeffrey Vedanayagam, Ching-Jung Lin, Alex S. Flynt, Jiayu Wen.

**Methodology:** Ranjith Papareddy, Michael Nodine.

**Project administration:** Eric C. Lai.

**Supervision:** Michael Nodine, Eric C. Lai.

**Writing – original draft:** Jeffrey Vedanayagam, Eric C. Lai.

**Writing – review & editing:** Jeffrey Vedanayagam, Eric C. Lai.

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
