## [Decision Letter · Decision Letter 0]

20 Mar 2023

Dear Dr Lai,

Thank you very much for submitting your Research Article entitled 'Regulatory logic of endogenous RNAi in silencing de novo genomic conflicts' to PLOS Genetics.

The manuscript was fully evaluated at the editorial level and by independent peer reviewers. The reviewers appreciated the attention to an important topic but identified some concerns that we ask you address in a revised manuscript.

We therefore ask you to modify the manuscript according to the review recommendations. Your revisions should address the specific points made by each reviewer.

Yours sincerely,

A. Aziz Aboobaker

Academic Editor

PLOS Genetics

Bret Payseur

Section Editor

PLOS Genetics

Reviewer's Responses to Questions

**Comments to the Authors:**

Reviewer #1: Overall- This is an interesting manuscript discussing a specific type of small RNA, known as hpRNA, which is dependent on the Drosophila dicer paralogue DCR-2. The authors characterise this type of RNA in Drosophila simulans dcr-2 mutants, demonstrating clear deregulation. They then characterise targets of hpRNAs and their rough “age” and make some speculations about how this may link to genomic conflicts, although these are not really backed up by any data. I think it shows some new insights into what sort of genes hpRNAs target and provides insights into the diversity of different types of small RNAs across evolution as hpRNAs seem to be restricted to arthropods. I have a few small comments about the analyses performed which the authors might want to consider (major points) and a few text suggestions (minor comments).

Major points

1) The analysis of the dysregulation of hpRNAs in dcr-2 mutants has some areas that I think could be improved.

i) RNAseq- it is immediately notable from Fig 1 A and B that there is a lot more variance in the changes that occur in simulans compared to melanogaster. This is likely technical rather than indicating increased number of dcr2 dependent transcripts: even miRNA precursors show much greater scatter in simulans. With this in mind, could the authors plot the Z score on the y axis instead- this will take into account the increased overall variance and would enable a clearer comparison of the extent to which hpRNAs are perturbed in the two species.

ii) Small RNA seq was normalized to spike in, which is great. However, the normalization was not very well described and I struggled to understand exactly what they did. The materials and methods describe normalizing to 52 sequences- is this normalizing separately to each and then averaging, or averaging across the 52 to get a single value for the spike in and then normalizing to this? The two are not equivalent.

Moreover, this normalization method will transform the data into a continuous distribution BEFORE normalization, which will lead to poor performance for low expression transcripts. An alternative would be to create a full counts table just for the spike transcripts and generate size factors via DESeq from this, and use this for normalization.

More information about the sRNAs would also be useful- could the authors please expand Figure 1 to contain MA plots for sRNAs and boxplots comparing fold change in miRNAs to fold change for hpRNAs.

2) I think that there is something missing from the logic that targets of youngest hpRNAs are the most dysregulated. If I am not mistaken this is based on finding the targets of the hpRNA derived siRNAs; however it is possible that some of these genes might be targeted by other siRNAs as well, which might also be dysregulated in the dcr-2 mutants. This could confound the conclusion because it might be the changes in the other small RNAs that is responsible for the changes in expression rather than the changes in hpRNA (i.e. these genes might not even be functional targets of hpRNA). I think this is unlikely to be a confounder but it would be good if the authors could rule it out. My suggestion would be to specifically select a subset of the targets of hpRNA that have sequence similarity to hpRNA but NO other small RNA (including miRNA seed matches). Then reproduce their age analysis on this subset. I think it may be enough data for them to reproduce their observation; if this now shows no clear difference then it might indicate that their hypothesis that hpRNA changes are responsible could be wrong or incomplete.

3) The conclusion that hpRNAs that are not shared in simulans and melanogaster are youngest is broadly true but in specific instances may be incorrect due to polymorphism in melanogaster meaning that some hpRNAs apparently missing in melanogaster are just not present in lab drosophila. It would be relatively straightforward for the authors to refine this set by searching the entire melanogaster collection for this subset of hpRNAs using blast and discarding the few that are in fact found in some dmel strains- I think this would be quite a good step to take.

Minor

All names of species should be written out in full. It’s quite difficult to follow otherwise, especially when many other species are introduced (D yakuba etc)

MA plot needs to be explained in the main text {p6}

Fig 1 E and F axis do not stretch far enough up.

Why is there no MA plot for the siRNAs? I think this would be worth including (see above).

“These were highly confident” p7 should be “these annotations were high confidence”

“and not simply by homology”- p8: didn’t make sense to me but I think the authors mean “and not simply by sequence complementarity to hpRNA derived siRNAs” unless I misunderstood.

“The "oldest" Dsim hpRNAs (i.e., still relatively young, but shared with Dmel)” is a very strange phrase; I suggest clarifying “D. simulans hpRNAs that were shared with D melanogaster

“broadens the scenario that the Dsim X” p13 not clear what this means

“described in detail about the existence” p 13 should be “described the existence…”

Reviewer #2: This manuscript presents evidence for an expanding network of novel hairpin RNAs in Drosophila simulans and speculates that these data reveal the importance of endogenous RNAi for intrachromosomal conflicts and speciation. The authors utilized mutants of Dicer-2 in both D. melanogaster and D. simulans to perform differential expression analysis of testis total RNA and testis short RNA sequencing libraries; in dcr-2 mutants, loss of siRNAs along with accumulation of their pri-hpRNAs and upregulation of genes containing sequences homologous to those siRNAs allowed them to identify hpRNA-target gene interactions. This analysis yielded only one novel hpRNA in D. melanogaster but revealed 21 novel hpRNA in D. simulans, and all 21 seem to be de novo acquired in the sim-clade (they are not found in closely related outgroups). Further, most of the D. simulans hpRNA target genes reside on the X chromosome, suggesting these pairs could be involved in a sex ratio distortion conflict.

Major Points:

1) The data and analyses support the argument that D. simulans has a "young" set of hpRNA-target interactions--with a bias for X-linked targets--that exhibit stronger repression relative to conserved "older" hpRNA-target pairs. However, no experimental evidence is provided to buttress the claim that any of the novel hpRNA-target pairs are de facto examples of a meiotic drive conflict. Based on the data presented, the authors could soften some of the language drawing conclusions about the connection between these newly identified hpRNAs and sex chromosome drive. Alternatively, the authors might produce additional data to support this argument. For example, a knockout of hp-88 (or hp-ballchen) could be made to test if a drive is unleashed.

2) The “connections to transposons” seems tenuous. Besides one hpRNA that may target BS2/Jockey, is there additional evidence for the connection to TE biology? Is BS2/Jockey upregulated in dcr-2 mutants? BS2/Jockey does not appear to be in Figure 6.

3) More generally, the structure of the writing can be improved. Multiple sections are difficult to follow for an audience not already very familiar with the specifics of this system. A few examples are highlighted below.

"Unexpected complexity in the Dsim hpRNA network related to Dox family loci"

First, referring to the inferred ancestral ODox as well as the contemporary putative first derivative of ODox both as ODox is confusing. Second, Figure 3B appears to illustrate hp-ODox1 and hp-ODox2 identically. A sequence alignment (or cartoon alignment) would be more helpful for the reader to understand their similarities and differences.

"Innovation of Dsim hpRNAs that target other HMG-box loci"

Is the argument being made here that HMG box factors are often targets of hpRNAs? If so, it would be helpful to state this clearly up front and briefly provide evidence for the claim; leave the explanations of the complexities of these hpRNAs for a later paragraph. Additionally, there is a cumbersome explanation of how tHMG is not in the MST-HMG subclass but is related protamine, which is possibly evolving rapidly, and the overall point of this opening paragraph is unclear. Although the complexity of the tHMG-related hairpins and targets it appreciated, it seems odd that the authors do not mention that both tHMG2-Xhet-1 and tHMG2-Xhet-2 are among the most highly up-regulated Dsim-specific targets (Fig. 6C). These data appear to highlight an intra-chromosomal conflict.

“Functional repression by hpRNAs is most overt for the youngest siRNA loci”

Some of these results may be overstated. For example, the increased expression of tapas is below the 1.5-fold threshold (Fig. 6D) yet is cited as supporting evidence for functional repression by hp-ODox1. Additionally, the increased expression of krimp is almost as high as increased expression of tapas despite the lack of siRNA targeting krimp, as noted previously (Fig. 3D and earlier in the text). Finally, is the comparison to miRNA appropriate?

Minor Points:

Page 4, last paragraph of Introduction: Is it surprising that young hpRNAs would have would have a strong effect in a conflict scenario?

Page 9, second paragraph: Typo “functiosn” should be functions.

Page 11, second paragraph: CG17358 does not agree with Figure 5B (CG17385). There are other instances of gene numbers in the text, tables, and figures not matching. Please check them for consistency.

Page 14, first paragraph: There may be a typo in “the miRNA pathway does not typically does adaptive targeting”.

Page 14: The explanation of protamines as they relate to genomic conflict may be more useful in the introduction. Additionally, it is not clear that data from this manuscript provides any new information in regards to piRNAs or TEs, so these paragraphs may not be relevant here.

Page 15, last paragraph: “…active genomic conflict scenarios that must be playing out in these species” should probably be worded as “may” be playing out in these species.

Page 16, last paragraph: This seems to be a typo “{Lin, 2018 #14350.”

Figure 3 caption: Another example of conflicting gene numbers: GD1739 or GD17329?

Figure 3 caption: (C) It is unclear what is being referenced with the text “and other regions of hp-ODox1 bear TE sequences.”

Figure 6 caption: “(D)” is missing.

Reviewer #3: This is a very nice overview of the dynamics of hpRNA evolution in the simulans clade. Using new Dicer alleles in simulans, the authors uncover a complex web of loci that interact through endogenous RNAi, presumably part of sex ratio conflict. I think the manuscript could be somewhat improved by providing, for figures 3,4 and 5 a) more context and b) alignments that support the models in figures 3,4 and 5. In particular, for figure 3, rather than simply presenting the model for the generation of Odox followed by other Dox variants and hp-ODox1,2, the model would be supported by combining phylogenetics. Otherwise, it is not clear if the model they propose is suppported. 3A and 3B are a models for the evolution of the Dox network, but it is not supported by data that is presented to the reader.

My only real concern is that many of the statements are not supported by statistical analysis. For example, they make the claim that new sim hpRNA targets are enriched on the X, but not new hpRNA loci. By my count, for figure 7, this is what they have for sim specific loci:

new hpRNAs: X: 6, Autosomes: 11

new targets: X: 11, Autosomes: 4

By a Fisher's exact, the p-value is 0.0416. So, this is significant enrichment. They should mention this.

Please perform additional statistical analyses when evaluating:

1. The enrichment of differentially expressed hpRNA loci in sim vs. mel based on the results of 2A and 2C.

2. The difference for fold expression change in 6C vs. 6B. The claim is that the new sim hpRNA targets are more sensitive to dicer loss, but this isn't statistically evaluated.

I have two other questions/requests.

1. Please provide an analysis of target/hpRNA %identity between 6C and 6B. Is there simply more sequence identity between sim specific hpRNAs and targets? Related, please provide quantitative data for the abundance of small RNAs that are antisense to each of the respective targets in 6C and 6B. There may be a quantitative difference that would be important to present to the reader.

2. Please provide a photo of the dicer mutant sim testes compared to wildtype and Dicer hets. It will be important for the reader to see these images to evaluate the results in 2C. If the testes are significantly atrophied, this might explain the greater variance in baseline log2 fold change between sim and mel.

**Have all data underlying the figures and results presented in the manuscript been provided?**

Reviewer #1: **No: **The authors haven't yet released these data.

Reviewer #2: Yes

Reviewer #3: Yes

PLOS authors have the option to publish the peer review history of their article (what does this mean?). If published, this will include your full peer review and any attached files.

Reviewer #1: No

Reviewer #2: No

Reviewer #3: No

---

## [Decision Letter · Decision Letter 1]

17 May 2023

Dear Dr Lai,

We are pleased to inform you that your manuscript entitled "Regulatory logic of endogenous RNAi in silencing de novo genomic conflicts" has been editorially accepted for publication in PLOS Genetics. Congratulations!

Yours sincerely,

A. Aziz Aboobaker

Academic Editor

PLOS Genetics

Bret Payseur

Section Editor

PLOS Genetics

Comments from the reviewers (if applicable):

This was very enjoyable paper to read.

Reviewer's Responses to Questions

**Comments to the Authors:**

Reviewer #1: I thank the reviewers for their considered responses. I’m satisfied that they have answered my questions very well. The only thing I would still request is that they should use full species names rather than abbreviations as I do think this would improve clarity for the casual reader.

**Have all data underlying the figures and results presented in the manuscript been provided?**

Reviewer #1: Yes

PLOS authors have the option to publish the peer review history of their article (what does this mean?). If published, this will include your full peer review and any attached files.

Reviewer #1: **Yes: **Peter Sarkies

**Data Deposition**

http://datadryad.org/submit?journalID=pgenetics&manu=PGENETICS-D-22-01459R1

**Press Queries**

---

## [Editor Report · Acceptance letter]

16 Jun 2023

PGENETICS-D-22-01459R1 

Regulatory logic of endogenous RNAi in silencing de novo genomic conflicts 

Dear Dr Lai, 

We are pleased to inform you that your manuscript entitled "Regulatory logic of endogenous RNAi in silencing de novo genomic conflicts" has been formally accepted for publication in PLOS Genetics! Your manuscript is now with our production department and you will be notified of the publication date in due course.

With kind regards,

Zsofi Zombor

PLOS Genetics

On behalf of:
